# Differentiable JPEG-based Input Perturbation for Knowledge Distillation Amplification via Conditional Mutual Information Maximization

**Siyu Chen**[*], **Kaixiang Zheng**[*], **Ahmed H. Salamah**[*] **& En-Hui Yang**
Department of Electrical and Computer Engineering, University of Waterloo
{s875chen, k56zheng, ahamsalamah, ehyang}@uwaterloo.ca

## Abstract

Maximizing conditional mutual information (CMI) has recently been shown to enhance the effectiveness of teacher networks in knowledge distillation (KD). Prior work achieves this by fine-tuning a pretrained teacher to maximize a proxy of its CMI. However, fine-tuning large-scale teachers is often impractical, and proxy-based optimization introduces inaccuracies. To overcome these limitations, we propose Differentiable JPEG-based Input Perturbation (DJIP), a plug-and-play framework that improves teacher–student knowledge transfer without modifying the teacher. DJIP employs a trainable differentiable JPEG layer inserted before the teacher to perturb teacher inputs in a way that directly increases CMI. We further introduce a novel alternating optimization algorithm to efficiently learn the coding parameters of the JPEG layer to maximize the perturbed CMI. Extensive experiments on CIFAR-100 and ImageNet, across diverse distillers and architectures, demonstrate that DJIP consistently improves student accuracy—achieving up to 4.11% gains—while remaining computationally lightweight and fully compatible with standard KD pipelines.

## 1 Introduction

Knowledge distillation (KD) (Buciluǎ et al., 2006; Hinton et al., 2015) has emerged as a pivotal technique for model compression, enabling the transfer of knowledge from large teacher models to lightweight student networks. This approach significantly improves student model performance without incurring high computational cost. Under resource constraints, KD is often simpler to apply and more robust in preserving accuracy compared to other compression techniques such as pruning (Sun et al., 2024a) and quantization (Lin et al., 2024), especially when deployment simplicity is a key concern.

Since the seminal work of Hinton et al. (2015), extensive research has sought to understand the underlying mechanisms of KD (Phuong & Lampert, 2021; Mobahi et al., 2020; Allen-Zhu & Li, 2023; Dao et al., 2021), and to develop more effective distillation techniques (Peng et al., 2019; Romero et al., 2014; Zhao et al., 2022; Zheng & YANG, 2024). However, in those conventional KD methods, the teacher model is typically trained solely to minimize its cross-entropy (CE) loss, with little attention paid to its ability to provide an informative supervision signal to student models.

To address this issue, student-oriented teacher (Cho & Hariharan, 2019; Wang et al., 2022; Tan & Liu, 2024; Yang et al., 2019; Dong et al., 2024; Ye et al., 2024; Hamidi et al., 2024) have been proposed to yield softer, more informative supervision signals. Notably, MCMI (Ye et al., 2024) demonstrates that maximizing conditional mutual information (CMI) during teacher training improves distillation effectiveness. However, these methods require modifying the teacher's weights, an impractical constraint in many real-world scenarios where the teacher is fixed or proprietary.

This retraining limitation has prompted an alternative line of research introducing perturbations at the input level. In conventional KD pipelines, both the teacher and student consume the same input

---

[*]Authors contributed equally.

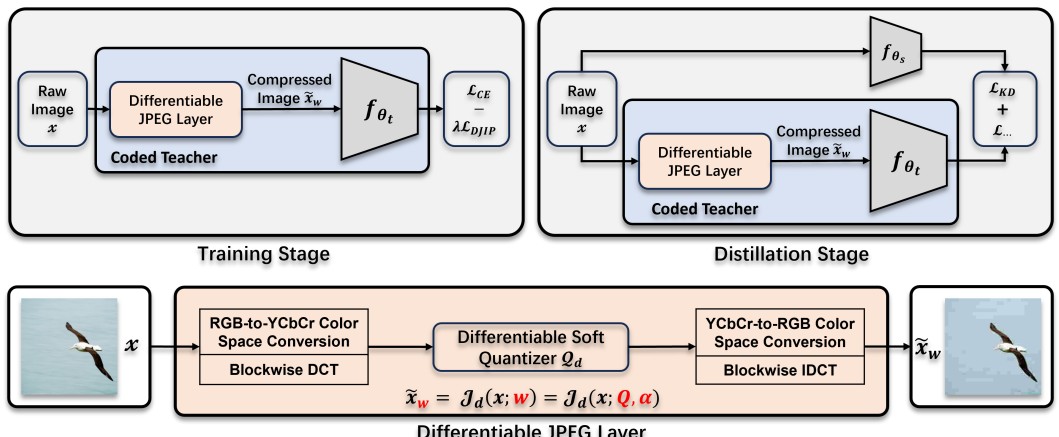

Figure 1: The overall training framework of DJIP is shown, centered around a differentiable JPEG coded teacher module (blue), which integrates a differentiable JPEG layer (orange). The process involves two stages: differentiable JPEG layer training and student distillation, with trainable parameters highlighted in red.

images, which may restrict the teacher's ability to transfer its full representational knowledge to the student. Recent works (Heo et al., 2018a; Nguyen-Duc et al., 2023; Zhang et al., 2021) suggest that feeding students with adversarial or divergent inputs generated by teachers can enhance the efficiency of knowledge transfer. Coded Knowledge Distillation (CKD) (Salamah et al., 2025a) further proposes that adaptively compressed input images can enhance the teacher's distillation efficacy. However, these approaches often incur high costs due to the need for generating additional samples, thereby increasing distillation complexity.

To overcome these limitations while further leveraging the benefits of input perturbation, we propose *Differentiable JPEG-based Input Perturbation* (DJIP). DJIP employs a trainable differentiable JPEG layer inserted before the teacher to perturb inputs in a way that directly increases CMI. The combination of the differentiable JPEG layer and the teacher is referred to as a differentiable JPEG *coded teacher*. This setup enables end-to-end learning of the coding parameters of the differentiable JPEG layer tailored to maximize the perturbed CMI of the teacher. Unlike MCMI, which fixes class-wise centroids during CMI maximization, potentially sacrificing the precision of the CMI proxy, we propose a novel alternating algorithm that reformulates the perturbed CMI maximization objective into a double minimization problem. This algorithm allows the centroids to be dynamically updated at each iteration, resulting in more stable and effective training.

To demonstrate the effectiveness of DJIP, we conduct extensive experiments on two datasets, covering both same- and cross-architecture distillation, and including both CNN and ViT models. Results consistently demonstrate the orthogonality of DJIP over various KD pipelines, including MCMI, and superiority over CKD. Specifically, DJIP can improve student Top-1 accuracy by up to 4.11%.

Our contributions are summarized as follows:

**Differentiable JPEG-based Input Perturbation:** We propose Differentiable JPEG-based Input Perturbation (DJIP), a plug-and-play framework that improves teacher–student knowledge transfer without modifying the teacher. DJIP employs a trainable differentiable JPEG layer inserted before the teacher to perturb teacher inputs in a way that directly increases CMI. During distillation, this framework can achieve significantly lower computational overhead while maintaining comparable functionality.

**Alternating Algorithm for Maximizing the Perturbed CMI:** We further introduce a novel alternating optimization algorithm to efficiently learn the coding parameters of the JPEG layer to maximize the perturbed CMI. The algorithm works by iteratively updating class centroids and the coding parameters.

**Comprehensive Empirical Evaluation:** We extensively evaluate DJIP across diverse datasets and model architectures, demonstrating its generalizability, consistency, effectiveness, orthogonality to existing methods, and compatibility with a wide range of knowledge distillation pipelines.

## 2 RELATED WORKS

A detailed review of related works, including conventional KD, KD with student-oriented teacher, and KD with input perturbation, is provided in Appendix A.1, where we also outline the most relevant directions for positioning our method and the advantages of our approach.

## 3 NOTATION AND PRELIMINARIES

### 3.1 NOTATION

Denote the i-th element of a vector $p$ as $p[i]$. For a positive integer $K$, let $[K] \triangleq \{1, \ldots, K\}$. For a multi-class classification task, assume that there are $C$ class labels with $[C]$ as the set of class labels. Also, a $C$-dimensional probability simplex is denoted by $\triangle^C$ for $C > 1$. The cross entropy of two probability distribution $P_1, P_2 \in \triangle^C$ is defined as $H(P_1, P_2) = \sum_{c=1}^{C} -P_1[c] \ln P_2[c]$, and their Kullback-Leibler (KL) divergence is defined as $D_{\mathrm{KL}}(P_1 \| P_2) = \sum_{c=1}^{C} P_1[c] \ln \frac{P_1[c]}{P_2[c]}$.

For any pair of random variables $(X, Y)$, denote its joint probability distribution by $P_{X,Y}(x, y)$ or simply $P(x, y)$ whenever there is no ambiguity, the marginal distribution of $Y$ by $P_Y(y)$, the conditional distribution of $Y$ given $X = x$ by $P_{Y|X}(\cdot \mid x)$, and the expected value with respect to $X$ by $\mathbb{E}_X(\cdot)$. The conditional mutual information (CMI) of $X$ and $Y$ given a third random variable $Z$ is $I(X; Y \mid Z) = H(X \mid Z) - H(X \mid Y, Z)$.

### 3.2 CONDITIONAL MUTUAL INFORMATION OF DNNS

A classification DNN with $C$ classes can be viewed as a mapping $x \mapsto f_\theta(x)$, where $x \in \mathbb{R}^d$ is an input image, $\theta$ denotes the model parameters, and $f_\theta(x) \in \triangle^C$ is the output class probability distribution in response to $x$. Whenever the context is clear, we omit $\theta$ and simply write $f(x)$. Let $\hat{y}$ denote the label predicted by the DNN in response to input $x$ with probability $f(x)[\hat{y}]$, and $y$ denote the ground-truth label of $x$. The set of distributions $f(x)$ in $\triangle^C$ corresponding to all input samples $x$ with the same ground-truth label $y$ forms a cluster in $\triangle^C$ (referred to as the $y$-$cluster$). Now let $(X, Y)$ be a pair of random variables representing a random input sample and its corresponding ground-truth label. Feed $X$ into the DNN and let $\hat{Y}$ denote the corresponding predicted label. As shown in Yang et al. (2025), $Y \to X \to \hat{Y}$ then forms a Markov chain with $P_{\hat{Y}|XY}(i \mid x, y) = f(x)[i]$, and the CMI between $X$ and $\hat{Y}$ given $Y = y$ can be computed as:

$$
\begin{aligned}
I(X; \hat{Y} \mid Y = y) &= \sum_x P_{X|Y}(x \mid y) \left[ \sum_{i=1}^{C} P_{\hat{Y}|XY}(\hat{Y} = i \mid x, y) \times \ln \frac{P_{\hat{Y}|XY}(\hat{Y} = i \mid x, y)}{P_{\hat{Y}|Y}(\hat{Y} = i \mid Y = y)} \right] \\
&= \mathbb{E}_{X|Y} \left[ D_{\mathrm{KL}}(f(X) \| S_y) \mid Y = y \right], \\
&\text{where } S_y = P_{\hat{Y}|Y}(\cdot \mid y) = \mathbb{E}_{X|Y}[f(X) \mid Y = y].
\end{aligned}
\tag{1}
$$

$I(X; \hat{Y} \mid Y = y)$ measures the concentration of the $y$-cluster, and $S_y$ can be viewed as the *centroid* of the $y$-cluster. Averaging over all such clusters, we obtain $I(X; \hat{Y} \mid Y)$, which reflects the average predictive concentration across all classes:

$$
I(X; \hat{Y} \mid Y) = \sum_{y \in [C]} P_Y(y) I(X; \hat{Y} \mid Y = y) = \mathbb{E}_{X,Y} \left[ D_{\mathrm{KL}}(f(X) \| S_Y) \right].
\tag{2}
$$

For a training set $\mathcal{D} = \{(x_i, y_i)\}_{i=1}^{N}$ drawn from an unknown distribution $P_{X,Y}$, we can approximate the CMI of model $f$ by its empirical value. To be specific, let $\mathcal{D}_y = \{x_j \in \mathcal{D} : y_j = y\}$. Denote the size of $\mathcal{D}_y$ by $|\mathcal{D}_y|$. The empirical values of CMI can be calculated as follows:

$$
I(X; \hat{Y} \mid Y) = \frac{1}{N} \sum_{y \in [C]} \sum_{x_j \in \mathcal{D}_y} D_{\mathrm{KL}}(f(x_j) \| S_y), \quad \text{where } S_y = \frac{1}{|\mathcal{D}_y|} \sum_{x_j \in \mathcal{D}_y} f(x_j), \text{ for } y \in [C].
\tag{3}
$$

### 3.3 DIFFERENTIABLE JPEG LAYER

JPEG (Pennebaker & Mitchell, 1992) is one of the most widely adopted lossy image compression standards in real-world applications. It achieves compression by exploiting spatial redundancy and perceptual irrelevance in natural images, resulting in high efficiency for storage and transmission. The standard JPEG pipeline first converts an RGB image $x$ into YCbCr, partitions it into non-overlapping $8\times8$ blocks, and applies the Discrete Cosine Transform (DCT). The resulting coefficients are quantized uniformly using tables $\boldsymbol{Q}$ and entropy-coded (e.g., Huffman coding). Reconstruction reverses this process via dequantization, inverse DCT, and conversion back to RGB. For simplicity, we denote the uniform quantizer as $\mathcal{Q}_u$.

To increase DNN nonlinearity, the JPEG-DL framework proposed by (Salamah et al., 2025b) introduces a differentiable JPEG layer into any underlying DNN. Specifically, JPEG-DL incorporates a novel differentiable JPEG layer as the input layer of the underlying DNN architecture. This layer simulates the standard JPEG codec but replaces the non-differentiable $\mathcal{Q}_u$ with a differentiable soft quantizer, denoted as $\mathcal{Q}_d$. This quantizer, parameterized by a quantization step size $q \in \boldsymbol{Q}$ and a sharpness parameter $\alpha \in \boldsymbol{\alpha}$, approximates $\mathcal{Q}_u$ via a smooth expectation over quantization bins, thereby enabling end-to-end gradient-based optimization. For simplicity, we denote the entire differentiable JPEG layer as $\mathcal{J}_d$, whose structure is illustrated in Fig. 1. Hence, the reconstructed image is given by $\tilde{x}_w = \mathcal{J}_d(x, w)$, where $w$ denotes the trainable parameters $(\boldsymbol{Q}, \boldsymbol{\alpha})$.

In JPEG-DL, the $\mathcal{J}_d$ is regarded as part of the overall DNN architecture. During training, the quantization (i.e., coding) parameters and the underlying DNN weights are jointly optimized to minimize the standard CE loss. In our work, however, the differentiable JPEG layer is used as a mechanism to perturb the input to the teacher and is separated from the teacher. Furthermore, during training, only the quantization parameters are optimized to maximize the perturbed CMI of the teacher with the teacher frozen completely.

## 4 METHODOLOGY

In this section, we first present the overall DJIP framework, then detail the objective function we use, and finally introduce our novel alternating optimization algorithm.

### 4.1 OVERALL FRAMEWORK

As illustrated in Figure 1, the overall DJIP framework consists of two stages:

**Differentiable JPEG Layer Training:** The JPEG layer first perturbs the input image $x$ into $\tilde{x}_w = \mathcal{J}_d(x, w)$, which is then fed to the teacher model. Under the objective function proposed in the next section, which balances the CE loss and the DJIP loss introduced later, the JPEG coding parameters are optimized to maximize the perturbed CMI.

**Student Distillation:** In this stage, the trained JPEG layer is integrated into the standard KD framework to perturb the teacher's input images. With its input perturbed by the trained JPEG layer to increase the perturbed CMI, the teacher can provide more informative supervision signals to student.

### 4.2 OBJECTIVE FUNCTION

With reference to Figure 1, given $\mathcal{J}_d$, $\tilde{X}_w$ is a deterministic function of $X$. Hence, the variables form a Markov chain $Y \rightarrow X \rightarrow \tilde{X}_w \rightarrow \hat{Y}$, which implies

$$I(X; \hat{Y} \mid Y) = I(X\tilde{X}_w; \hat{Y} \mid Y) = I(\tilde{X}_w; \hat{Y} \mid Y). \tag{4}$$

We refer to $I(\tilde{X}_w; \hat{Y} \mid Y)$ as the perturbed CMI. Our goal is to perturb the input $X$ such that the perturbed CMI is maximized to a certain extent, while simultaneously minimizing the CE loss with respect to the ground-truth labels. Following the spirit of Ye et al. (2024), we formulate our CE–CMI optimization problem over $w$ as

$$\min_w \left\{ \mathbb{E}_X \left[ H(P_{Y|X}, f(\tilde{X}_w)) \right] - \lambda \, I(\tilde{X}_w; \hat{Y} \mid Y) \right\}, \tag{5}$$

where $\lambda > 0$ is a hyper-parameter that balances the CE-CMI trade-off. In contrast to Ye et al. (2024), where optimization is carried out over the pretrained model parameters $\theta$, the DJIP framework freezes the teacher and updates only the JPEG parameters $w$.

However, as stated by Ye et al. (2024), maximizing $I(X; \hat{Y} \mid Y)$ poses certain challenges because the term $S_y$ depends on $f(x_j), \forall x_j \in \mathcal{D}_y$ (see Equation 3) which is not well-suited for numerical solutions and cannot be efficiently parallelized on GPUs. The same challenges apply to the optimization problem in Equation 5 as well. Ye et al. (2024) circumvents this issue by fixing the centroids $S_y$ obtained from a pretrained model. Nevertheless, these centroids may shift during fine-tuning, so using fixed centroids may partially mitigate the issue, but does not fully resolve the theoretical limitations of their method.

To address the above challenges theoretically, we introduce a dummy "backward channel" and reformulate the optimization problem in Equation 5 as a double minimization problem, as shown in the following theorem and proved in Appendix A.2:

**Theorem 1** *For any $i, y \in [C]$, let $Q(\cdot \mid i, y)$ denote a dummy conditional distribution over the input space $X$ ("backward channel"). Then for any $\lambda > 0$,*

$$\min_w \left\{ \mathbb{E}_X \left[ H(P_{Y|X}, f(\tilde{X}_w)) \right] - \lambda I(\tilde{X}_w; \hat{Y} \mid Y) \right\}$$

$$\equiv \min_w \min_{\{Q(\cdot|i,y)\}} \left\{ \sum_x P(x) H(P_{Y|X}(\cdot \mid x), f(\tilde{x}_w)) - \lambda \sum_{x,y} P(x,y) \sum_{i=1}^C f(\tilde{x}_w)[i] \ln Q(x \mid i, y) \right\}, \quad (6)$$

*where the inner minimization above is achieved when*

$$Q(x \mid i, y) = \frac{P_{X|Y}(x \mid y) f(\tilde{x}_w)[i]}{P_{\hat{Y}|Y}(i \mid y)}. \quad (7)$$

In practice, the training set is randomly partitioned into $B$ mini-batches $\mathcal{B}^b$ for $b \in [B]$, each of size $|\mathcal{B}|$. When the joint distribution of $(X, Y)$ is unknown, the objective function can be approximated by its empirical estimate over a mini-batch $\mathcal{B}$. Accordingly, based on Theorem 1, the empirical objective function $\mathcal{L}_{\text{emp}}$ for the proposed DJIP method can be formulated as:

$$\mathcal{L}_\mathcal{B}(\lambda, w, \{Q(\cdot|i,y)\}) = \underbrace{\frac{1}{|\mathcal{B}|} \sum_{(x,y) \in \mathcal{B}} (-\ln f(\tilde{x}_w)[y])}_{\mathcal{L}_{\text{CE}}} - \lambda \underbrace{\frac{1}{|\mathcal{B}|} \sum_{(x,y) \in \mathcal{B}} \left[ \sum_{i=1}^C f(\tilde{x}_w)[i] \ln Q(x|i,y) \right]}_{\mathcal{L}_{\text{DJIP}}}. \quad (8)$$

### 4.3 AN ALTERNATING CMI MAXIMIZATION ALGORITHM

Based on equation 6 to equation 8, we are now ready to present our algorithm for solving the optimization problem in equation 5, which optimizes $w$ and $\{Q(\cdot \mid i, y)\}$ alternatively to minimize the objective function in equation 8:

**Step 1:** Fix $w$, let $\mathcal{D}_{x,y} = \{(x_j, y_j) \in \mathcal{D} : x_j = x, y_j = y\}$. According to 7, $\{Q(\cdot \mid i, y)\}$ can be updated in two steps. (1) update the centroids $S_y$ empirically according to Equation 3; and (2) calculate the empirical version of $Q(x \mid i, y)$ according to Equation 7:

$$S_y[i] = P_{\hat{Y}|Y}(i \mid y) = \frac{1}{|\mathcal{D}_y|} \sum_{x_j \in \mathcal{D}_y} f(\mathcal{J}_d(x_j, w))[i], \quad \forall i, y \in [C], \quad (9)$$

$$Q(x \mid i, y) = \frac{P_{X|Y}(x \mid y) f(\mathcal{J}_d(x, w))[i]}{P_{\hat{Y}|Y}(i \mid y)} = \frac{\frac{|\mathcal{D}_{x,y}|}{|\mathcal{D}_y|} f(\mathcal{J}_d(x, w))[i]}{P_{\hat{Y}|Y}(i \mid y)}, \quad \forall (x,y) \in \mathcal{D}, i \in [C]. \quad (10)$$

**Step 2:** Fix $\{Q(\cdot \mid i, y)\}$, $w$ can be updated using a standard deep learning process through stochastic gradient descent (SGD).

A detailed pseudo-code of the alternating optimization algorithm is provided in Appendix A.3.

## 5 EXPERIMENTS

**Terminologies.** To evaluate the performance of DJIP, we conduct a series of experiments. This section presents the main experimental results and demonstrates the extent to which DJIP improves

accuracy over state-of-the-art KD methods. For clarity, we denote the teachers trained solely with CE loss, with MCMI estimator (Ye et al., 2024), and with the proposed DJIP method as the *CE* teacher, *MCMI* teacher, and *DJIP* teacher, respectively.

**Plug-and-Play Nature.** In all experiments reported, the JPEG layer functions as a lens to facilitate improved teacher distillation. Once removed, the model reverts to its conventional form without any residual effect. Furthermore, no hyperparameters of the underlying KD methods are modified; all configurations remain identical to those used in the original benchmark settings.

We conduct extensive experiments on ImageNet and CIFAR-100 with diverse model architectures. Moreover, we show that DJIP can be effectively applied to cross-paradigm distillation between CNNs and Vision Transformers (ViTs). The results further demonstrate that DJIP is complementary to existing techniques and remains orthogonal to the latest state-of-the-art benchmarks.

## 5.1 CIFAR-100 RESULTS

The CIFAR-100 dataset is a widely used benchmark for image classification, comprising 60,000 color images with a resolution of $32 \times 32$ pixels, categorized into 100 classes. Following the experimental setup of Tian et al. (2019), we conduct experiments with 7 teacher-student pairs sharing identical architectures (see Table 1) and 6 pairs using different architectures (see Table 2). Each experiment is repeated across three independent runs, and the average accuracy is reported.

For a comprehensive comparison, we evaluate our DJIP teacher against the conventional CE teacher using state-of-the-art distillation methods. These include logit-based approaches: KD (Hinton et al., 2015), DKD (Zhao et al., 2022), DIST (Huang et al., 2022), and WTTM (Zheng & YANG, 2024); relation-based approaches: CC (Peng et al., 2019) and RKD (Park et al., 2019); and feature-based approaches: AT (Zagoruyko & Komodakis, 2016), FitNet (Romero et al., 2014), FT (Kim et al., 2020), SP (Tung & Mori, 2019), ITRD (Miles et al., 2021), CRD (Tian et al., 2019), and LSKD (Sun et al., 2024b). Methods of the same category are grouped in all tables. Details of the training setups, including the training of DJIP teachers, student distillation procedures, the choice of hyperparameters, and the visualization of the optimized 128 quantization parameters are provided in Appendix A.5 and A.10.

For both CE and DJIP teachers, we report their CMI values measured on the training set without data augmentation. As shown in Tables 1 and 2, replacing the CE teacher with the DJIP teacher consistently improves the student performance, regardless of whether the teacher and student architectures are the same. These improvements are observed across all evaluated methods, with accuracy gains of up to 2.44%. Notably, the improvements are more pronounced when the teacher and student architectures differ, i.e., when there exists a larger capacity gap between them.

## 5.2 IMAGENET RESULTS

ImageNet is a large-scale dataset used in visual classification tasks, containing approximately 1.2 million training images and 50,000 validation images. Following the implementation of Zhao et al. (2022), we conduct experiments on two widely used teacher-student pairs (see Table 3) and six representative distillation methods: KD (Hinton et al., 2015), AT (Zagoruyko & Komodakis, 2016), DKD (Zhao et al., 2022), LSKD (Sun et al., 2024b), WSLD (Zhou et al., 2021), and ReviewKD (Chen et al., 2021).

Across all knowledge transfer methods reported in Table 3, we observe that replacing the CE teacher with the DJIP teacher consistently improves the student Top-1 accuracy as well. Details on the training setups, including training of DJIP teachers, student distillation, and the choice of hyperparameters, as well as an analysis on the effect of hyperparameters for DJIP teachers, are provided in Appendix A.5 and A.6.

## 5.3 VISION TRANSFORMER RESULTS

In previous experiments on CIFAR-100, we demonstrated the effectiveness of DJIP for distillation between CNN architectures, whether identical or different. In this section, following Hao et al. (2023); Li et al. (2022), we extend DJIP to address the challenge of cross-paradigm distillation between CNNs and ViTs on CIFAR-100. The experimental results are summarized in Table 4. We

Table 1: The test accuracy (%) of students on CIFAR-100 (averaged over 3 runs), with teacher-student pairs in the same architecture. We use asterisk (*) to identify the results reproduced on our local machines. The small print denotes the improvement achieved by using the DJIP teacher.

| Teacher Acc | WRN-40-2 75.61 | | WRN-40-2 75.61 | | ResNet-56 72.34 | | ResNet-110 74.31 | | ResNet-110 74.31 | | VGG-13 74.64 | | ResNet-32×4 79.41 | |
|---|---|---|---|---|---|---|---|---|---|---|---|---|---|---|
| Student Acc | WRN-16-2 73.26 | | WRN-40-1 71.98 | | ResNet-20 69.06 | | ResNet-20 69.06 | | ResNet-32 71.14 | | VGG-8 70.36 | | ResNet-8×4 72.50 | |
| | CE | DJIP | CE | DJIP | CE | DJIP | CE | DJIP | CE | DJIP | CE | DJIP | CE | DJIP |
| CMI | 0.026 | 0.501 | 0.026 | 0.501 | 0.158 | 0.724 | 0.061 | 0.565 | 0.061 | 0.565 | 0.015 | 0.252 | 0.006 | 0.276 |
| KD | 74.92 | 75.64 +0.72 | 73.54 | 74.41 +0.87 | 70.66 | 71.20 +0.54 | 70.67 | 71.65 +0.98 | 73.08 | 73.71 +0.63 | 72.98 | 74.01 +1.03 | 73.33 | 74.38 +1.05 |
| DKD | 75.63* | 76.08 +0.45 | 74.85 | 75.14 +0.29 | 71.58* | 71.86 +0.28 | 71.51 | 71.72 +0.21 | 74.11 | 74.22 +0.11 | 74.68 | 74.93 +0.25 | 76.32 | 76.55 +0.23 |
| DIST | 75.51 | 75.98 +0.47 | 74.26 | 74.99 +0.73 | 71.75 | 71.97 +0.22 | 71.65 | 71.90 +0.25 | 73.69 | 73.90 +0.21 | 73.89 | 74.31 +0.42 | 76.31 | 76.60 +0.29 |
| WTTM | 76.37 | 76.70 +0.33 | 74.58 | 74.98 +0.40 | 71.92 | 72.15 +0.23 | 71.67 | 71.90 +0.23 | 74.13 | 74.32 +0.19 | 74.44 | 74.81 +0.37 | 76.06 | 76.61 +0.55 |
| CC | 73.56 | 73.80 +0.24 | 72.21 | 72.48 +0.27 | 69.63 | 69.93 +0.30 | 69.48 | 69.87 +0.39 | 71.48 | 71.86 +0.38 | 70.71 | 71.12 +0.41 | 72.97 | 73.23 +0.26 |
| RKD | 73.35 | 74.09 +0.74 | 72.22 | 72.36 +0.14 | 69.61 | 70.19 +0.58 | 69.25 | 69.85 +0.60 | 71.82 | 72.45 +0.63 | 71.48 | 71.84 +0.36 | 71.90 | 72.69 +0.79 |
| AT | 74.08 | 74.51 +0.43 | 72.77 | 73.22 +0.45 | 70.55 | 70.81 +0.26 | 70.22 | 70.63 0.41 | 72.31 | 72.84 +0.53 | 71.43 | 72.03 +0.60 | 73.44 | 73.94 +0.50 |
| FitNet | 73.58 | 74.20 +0.62 | 72.24 | 72.84 +0.60 | 69.21 | 69.83 +0.62 | 68.99 | 69.40 +0.41 | 71.06 | 71.42 +0.36 | 71.02 | 71.92 +0.90 | 73.50 | 73.83 +0.33 |
| FT | 73.25 | 73.55 +0.30 | 71.59 | 71.93 +0.34 | 69.84 | 70.20 +0.36 | 70.22 | 70.70 +0.48 | 72.37 | 72.54 +0.17 | 70.58 | 71.28 +0.70 | 72.86 | 73.76 +0.90 |
| SP | 73.83 | 74.35 +0.52 | 72.43 | 72.99 +0.56 | 69.67 | 70.83 +1.16 | 70.04 | 71.05 +1.01 | 72.69 | 73.37 +0.68 | 72.68 | 73.42 +0.74 | 72.94 | 73.62 +0.68 |
| ITRD | 76.12 | 76.33 +0.21 | 75.18 | 75.23 +0.05 | 71.26* | 71.44 +0.18 | 71.52* | 71.86 +0.34 | 74.26 | 74.30 +0.04 | 74.86 | 75.00 +0.14 | 76.19 | 76.28 +0.09 |
| CRD | 75.48 | 76.01 +0.53 | 74.14 | 74.57 +0.43 | 71.16 | 71.61 +0.45 | 71.46 | 71.79 +0.33 | 73.48 | 73.92 +0.44 | 73.94 | 74.35 +0.41 | 75.51 | 75.85 +0.34 |
| LSKD | 76.11 | 76.35 +0.24 | 74.37 | 74.89 +0.52 | 71.26* | 71.39 +0.13 | 71.48 | 71.60 +0.12 | 73.67* | 73.92 +0.25 | 74.36 | 74.90 +0.54 | 76.62 | 76.96 +0.34 |

Table 2: The test accuracy (%) of students on CIFAR-100 (averaged over 3 runs), with teacher-student pairs in different architectures.

| Teacher Acc | ResNet-50 79.34 | | ResNet-50 79.34 | | ResNet-32×4 79.41 | | ResNet-32×4 79.41 | | WRN-40-2 75.61 | | VGG-13 74.64 | |
|---|---|---|---|---|---|---|---|---|---|---|---|---|
| Student Acc | MobileNetV2 64.60 | | VGG-8 70.36 | | ShuffleNetV1 70.50 | | ShuffleNetV2 71.82 | | ShuffleNetV1 70.50 | | MobileNetV2 64.60 | |
| Method | CE | DJIP | CE | DJIP | CE | DJIP | CE | DJIP | CE | DJIP | CE | DJIP |
| CMI | 0.009 | 0.341 | 0.009 | 0.341 | 0.006 | 0.276 | 0.006 | 0.276 | 0.026 | 0.501 | 0.015 | 0.252 |
| KD | 67.35 | 69.50 +2.15 | 73.81 | 74.48 +0.67 | 74.07 | 75.64 +1.57 | 74.45 | 76.24 +1.79 | 74.83 | 76.36 +1.53 | 67.37 | 68.87 +1.50 |
| DKD | 70.35 | 71.18 +0.83 | 73.94 | 75.87 +1.93 | 76.45 | 77.24 +0.79 | 77.07 | 77.52 +0.45 | 76.70 | 77.16 +0.46 | 69.71 | 70.20 +0.49 |
| DIST | 68.66 | 69.59 +0.93 | 74.11 | 74.80 +0.69 | 76.34 | 76.64 +0.30 | 77.35 | 77.95 +0.60 | 76.22 | 76.69 +0.47 | 68.50 | 69.01 +0.51 |
| WTTM | 69.59 | 69.98 +0.39 | 74.82 | 75.19 +0.37 | 74.37 | 74.95 +0.58 | 76.55 | 77.30 +0.75 | 75.42 | 76.24 +0.82 | 69.16 | 69.41 +0.25 |
| CC | 65.43 | 65.79 +0.36 | 70.25 | 70.92 +0.67 | 71.14 | 72.17 +1.03 | 71.29 | 73.25 +1.96 | 71.38 | 72.22 +0.84 | 64.86 | 65.83 +0.97 |
| RKD | 64.43 | 65.56 +1.13 | 71.50 | 71.93 +0.43 | 72.28 | 73.40 +1.12 | 73.21 | 74.27 +1.06 | 72.21 | 73.85 +1.64 | 64.52 | 66.04 +1.52 |
| AT | 58.58 | 60.02 +1.44 | 71.84 | 72.42 +0.58 | 71.73 | 73.93 +2.20 | 72.73 | 74.16 +1.43 | 73.32 | 75.27 +1.95 | 59.40 | 61.24 +1.84 |
| FitNet | 63.16 | 64.05 +0.89 | 69.39* | 69.57 +0.18 | 73.59 | 74.47 +0.88 | 73.54 | 74.65 +1.11 | 73.73 | 74.02 +0.29 | 64.14 | 65.46 +1.32 |
| FT | 60.99 | 62.82 +1.83 | 70.29 | 71.24 +0.95 | 71.75 | 73.41 +1.66 | 72.50 | 73.80 +1.30 | 72.03 | 73.80 +1.77 | 61.78 | 62.57 +0.79 |
| SP | 68.08 | 68.58 +0.50 | 73.34 | 73.87 +0.53 | 73.48 | 75.92 +2.44 | 74.56 | 76.28 +1.72 | 74.52 | 76.32 +1.80 | 66.30 | 67.68 +1.38 |
| ITRD | 71.34 | 72.06 +0.72 | 75.49 | 75.91 +0.42 | 76.91 | 77.30 +0.39 | 77.40 | 77.92 +0.52 | 77.09 | 77.31 +0.22 | 70.39 | 70.91 +0.52 |
| CRD | 69.11 | 70.01 +0.90 | 74.30 | 74.55 +0.25 | 75.11 | 75.79 +0.68 | 75.65 | 76.32 +0.67 | 76.05 | 76.21 +0.16 | 69.73 | 69.89 +0.16 |
| LSKD | 69.02 | 70.39 +1.37 | 74.88* | 75.21 +0.33 | 75.67* | 76.38 +0.71 | 75.56 | 77.12 +1.56 | 76.56* | 76.78 +0.22 | 68.61 | 69.93 +1.32 |

Table 3: The test accuracy (%) of students on ImageNet.

| Teacher | Student | Method | CMI | KD | AT | DKD | LSKD | WSLD | ReviewKD |
|---|---|---|---|---|---|---|---|---|---|
| ResNet-34 73.31 | ResNet-18 69.76 | CE | 0.720 | 70.66 | 70.70 | 71.70 | 71.42 | 71.73 | 71.61 |
| | | DJIP | 0.738 | 71.65 | 70.78 | 72.08 | 71.65 | 71.87 | 71.72 |
| | | Δ | / | +0.99 | +0.08 | +0.38 | +0.23 | +0.14 | +0.11 |
| ResNet-50 76.16 | MobileNetV1 68.87 | CE | 0.600 | 70.50 | 69.56 | 72.05 | 72.18 | 72.02 | 72.56 |
| | | DJIP | 0.649 | 70.92 | 70.57 | 72.52 | 72.37 | 72.78 | 73.04 |
| | | Δ | / | +0.42 | +1.01 | +0.47 | +0.19 | +0.76 | +0.48 |

include one teacher-student pair for each paradigm setting: ViT-to-CNN, CNN-to-ViT, and ViT-to-ViT. The results clearly show that our proposed method is also effective for knowledge transfer across heterogeneous architectural paradigms and highlight the potential of DJIP in distilling ViTs.

Table 4: The test accuracy (%) of students on CIFAR-100 (averaged over 3 runs), with teacher-student pairs in heterogeneous architectural paradigms.

| Teacher | Student | Method | CMI | KD | DIST | DKD | CC | RKD | CRD |
|---|---|---|---|---|---|---|---|---|---|
| ViT-S 92.04 | ResNet-18 74.01 | CE | 0.100 | 77.26 | 76.49 | 78.10 | 74.26 | 73.72 | 76.60 |
| | | DJIP | 0.171 | 79.06 | 78.04 | 80.15 | 74.87 | 75.85 | 77.75 |
| | | Δ | / | +1.80 | +1.55 | +2.05 | +0.61 | +2.13 | +1.15 |
| ConvNeXt-T 88.41 | DeiT-T 68.00 | CE | 0.229 | 72.99 | 73.55 | 74.60 | 68.01 | 69.79 | 65.94 |
| | | DJIP | 0.256 | 75.00 | 74.70 | 75.57 | 69.60 | 70.38 | 66.63 |
| | | Δ | / | +2.01 | +1.15 | +0.97 | +1.59 | +0.59 | +0.69 |
| ViT-S 92.04 | DeiT-T 68.00 | CE | 0.100 | 69.86 | 70.57 | 71.41 | 68.62 | 69.39 | 65.46 |
| | | DJIP | 0.171 | 73.97 | 73.96 | 74.90 | 69.86 | 70.11 | 65.75 |
| | | Δ | / | +4.11 | +3.39 | +3.49 | +1.24 | +0.72 | +0.29 |

# 6 ANALYSIS

## 6.1 ABLATION STUDY

As discussed in the contributions outlined in Section 1, DJIP introduces two key components: (1) a differentiable JPEG layer, and (2) an alternating CMI maximization algorithm. In this section, we conduct an ablation study to isolate and analyze the effectiveness of each component.

As shown in Table 5, when the differentiable JPEG layer is added but the fixed-centroid method from Ye et al. (2024) is retained (referred to as 'JMCMI'), the resulting method generally outperforms various KD baselines. Furthermore, when the alternating maximization algorithm is additionally applied, resulting in our proposed DJIP, the performance is further improved over JMCMI in general.

In summary, both the differentiable JPEG layer and the alternating CMI maximization algorithm contribute positively to the overall performance. Each component individually enhances the KD process, and their combined use in DJIP yields the best results, demonstrating a synergistic effect.

Table 5: The test accuracy (%) of students on CIFAR-100 (averaged over 3 runs), with teacher-student pairs of the same- and different-architecture.

| Teacher | Student | Method | CMI | KD | DKD | DIST | CC | RKD | AT | FitNet | FT | SP | ITRD | CRD |
|---|---|---|---|---|---|---|---|---|---|---|---|---|---|---|
| WRN-40-2 | WRN-16-2 | CE | 0.026 | 74.92 | 75.63 | 75.51 | 73.56 | 73.35 | 74.08 | 73.58 | 73.25 | 73.83 | 76.12 | 75.48 |
| | | JMCMI | 0.505 | 75.53 | 75.81 | 75.83 | 73.62 | 73.74 | 74.28 | 73.75 | 73.39 | 74.12 | 75.93 | 76.39 |
| | | DJIP | 0.501 | 75.64 | 76.08 | 75.98 | 73.80 | 74.09 | 74.51 | 74.20 | 73.55 | 74.35 | 76.33 | 76.01 |
| WRN-40-2 | ShuffleNetV1 | CE | 0.026 | 74.83 | 76.70 | 76.22 | 71.38 | 72.21 | 73.32 | 73.73 | 72.03 | 74.52 | 77.09 | 76.05 |
| | | JMCMI | 0.505 | 75.95 | 76.56 | 76.85 | 71.92 | 73.39 | 75.10 | 73.81 | 73.25 | 75.96 | 77.14 | 76.14 |
| | | DJIP | 0.501 | 76.36 | 77.16 | 76.69 | 72.22 | 73.85 | 75.27 | 74.02 | 73.80 | 76.32 | 77.31 | 76.21 |

## 6.2 ORTHOGONALITY OVER MCMI

Since DJIP shares MCMI's goal of maximizing the teacher's CMI, we examine whether our method is orthogonal to MCMI and can further improve performance. Following the experimental setup of (Ye et al., 2024), we prepend our trainable JPEG layer to the MCMI teacher to obtain the DJIP–MCMI teacher reported in Table 6.

The results in Table 6 indicate that applying our DJIP method yields additional improvements in Top-1 accuracy, with gains of up to 0.92% absolute. These findings suggest that DJIP is not only complementary but also orthogonal and additive to MCMI, as it explicitly optimizes the input space, which is not considered in MCMI.

Table 6: The test accuracy (%) of students on CIFAR-100 (averaged over three runs), with teacher-student pairs of the same- and different-architecture.

| Teacher | Student | Method | CMI | KD | DKD | SP | CRD | RKD |
|---|---|---|---|---|---|---|---|---|
| VGG-13 74.64 | VGG-8 70.36 | MCMI | 0.1298 | 73.83 | 74.87 | 73.29 | 74.23 | 72.03 |
| | | DJIP-MCMI | 0.1402 | 74.26 | 74.96 | 74.21 | 74.60 | 72.36 |
| | | Δ | / | +0.43 | +0.09 | +0.92 | +0.37 | +0.33 |
| VGG-13 74.64 | MobileNetV2 64.60 | MCMI | 0.1298 | 69.14 | 70.35 | 67.83 | 69.98 | 65.37 |
| | | DJIP-MCMI | 0.1402 | 69.40 | 70.51 | 68.33 | 70.46 | 65.90 |
| | | Δ | / | +0.26 | +0.16 | +0.50 | +0.48 | +0.53 |

## 6.3 COMPARISON WITH MCMI

As mentioned in Section 4, our DJIP method addresses the fixed-centroid problem of MCMI by introducing an alternating optimization algorithm. Consequently, when compared with the CIFAR-100 and ImageNet results[1] of MCMI applied to the same KD methods, our method achieves comparable or better performance across most cases, as presented in Figure 2.

It is important to note that MCMI has a much larger degree of freedom, as it optimizes the parameters of the pretrained model, whereas our method only adjusts the 128 quantization parameters of the JPEG layer. Despite this limitation, our method still matches or outperforms MCMI in multiple scenarios, demonstrating the effectiveness of learning solely through compression parameters.

## 6.4 COMPARISON WITH CKD AND TALD

As discussed in Section 2, our DJIP method is capable of exploring a significantly larger continuous quantization space than CKD. Consequently, when compared with the CIFAR-100 and ImageNet results[2] of CKD applied to the same KD methods, our approach consistently achieves comparable or superior performance. A brief comparison of CIFAR-100 experimental results is shown in Figure 2.

It is worth noting that CKD adaptively selects the optimal quantization table for each input image; that is, two different images may be compressed using different tables chosen to maximize teacher effectiveness, which is computationally expensive. In contrast, our method employs a fixed quantization table shared across all input images throughout the entire distillation process. Moreover, since CKD achieves comparable performance to another input-perturbed-based method, TALD (Nguyen-Duc et al., 2023), we also include TALD for comparison in Figure 2.

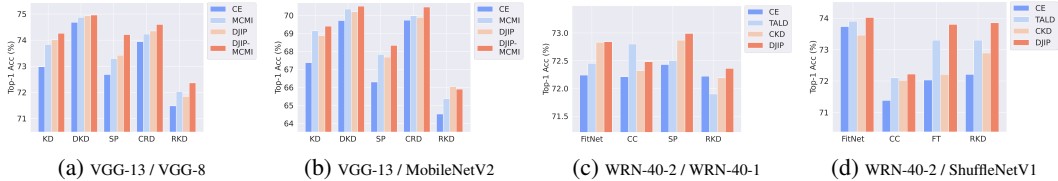

(a) VGG-13 / VGG-8    (b) VGG-13 / MobileNetV2    (c) WRN-40-2 / WRN-40-1    (d) WRN-40-2 / ShuffleNetV1

Figure 2: (a), (b) Comparison with MCMI under different student architectures on CIFAR-100. (c), (d) Comparison with CKD and TALD under different student architectures on CIFAR-100.

For further analysis, we refer the reader to Appendix A.10 and A.11, which provide visualizations of the learned quantization step sizes and the output probability distributions.

## 7 CONCLUSION

In this work, we have introduced Differentiable JPEG-based Input Perturbation (DJIP), a novel framework that enhances the transferability of a fixed teacher by incorporating a differentiable JPEG compression layer. By jointly optimizing CE loss and CMI values through a tailored alternating algorithm, our method enables the teacher to produce more informative supervision signals without modifying its original weights. Extensive experiments on ImageNet and CIFAR-100, across both CNNs and ViTs architectures, demonstrate that DJIP consistently improves student accuracy (up to 4.11% over standard baselines). Comprehensive analyses further confirm the effectiveness and stability of our alternating CMI optimization. Beyond empirical gains, DJIP offers a lightweight and generalizable mechanism for enhancing knowledge distillation by exploiting input-space perturbations using trainable quantization. Since the teacher remains unchanged, DJIP is particularly suitable for scenarios with strict deployment or integrity constraints.

---

[1]Please refer to Table 2 of Ye et al. (2024)

[2]Please refer to Table 3 of Salamah et al. (2025a)

## REPRODUCIBILITY STATEMENT

We have made significant efforts to ensure the reproducibility of our work. All implementation details of the proposed method, including model architectures, training procedures, as well as detailed descriptions of data preprocessing steps and hyperparameter settings, are provided in the appendix. To further facilitate reproducibility, we include executable source code and usage guidelines in the supplementary materials.

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

# A  APPENDIX

## A.1  RELATED WORKS

**Conventional KD:** Also known as knowledge transfer, KD is a model compression technique in which a compact student model learns to mimic the behavior of a larger teacher model. This concept was first introduced by Buciluǎ et al. (2006), who trained a smaller model to match the logits of a larger one. Hinton et al. (2015) later popularized KD by introducing temperature-based softening to both teacher and student logits. Since then, KD methods have proliferated and can be broadly categorized into three groups according to Yang et al. (2023): (i) response/logit-based methods (Hinton et al., 2015; Heo et al., 2018a; Zheng & YANG, 2024; Hamidi, 2024; Lan et al., 2018; Stanton et al., 2021; Chen et al., 2019; Li et al., 2020; Beyer et al., 2022; Zhao et al., 2022; Miles et al., 2021; Sun et al., 2024b; Huang et al., 2022); (ii) feature-based methods (Romero et al., 2014; Zagoruyko & Komodakis, 2016; Yim et al., 2017; Chen et al., 2021; Yang et al., 2021; Kim et al., 2020; Heo et al., 2018b; Tung & Mori, 2019; Tian et al., 2019); (iii) relation-based methods (Park et al., 2019; Peng et al., 2019; Liu et al., 2019; Yang et al., 2022). Building upon these conventional KD frameworks, which aim to improve the student model's ability to mimic the teacher's logits, intermediate features, or inter-feature relations, two main orthogonal directions have emerged to further enhance student performance.

**KD with Student-Oriented Teacher:** Rather than using teachers who only care about their own performance, several works aim to find or train teachers better suited to the students. Cho & Hariharan (2019); Wang et al. (2022) show that early stopping during teacher training or using earlier checkpoints preserves higher mutual information between inputs and outputs, providing more informative soft targets. Tan & Liu (2024); Yang et al. (2019) encourage teachers to produce more dispersed probability distributions via auxiliary losses, while Dong et al. (2024) shows that explicitly imposing the Lipschitz and consistency constraint in teacher training can facilitate the learning of the true label distribution and thus improve the student performance.

Meanwhile, Ye et al. (2024) demonstrates that training teachers with the maximum CMI (MCMI) estimator, rather than the conventional minimum CE objective, yields better approximations of the true Bayes conditional probability distribution (BCPD) of label $y$ given input $x$, thereby significantly enhancing student performance by capturing richer contextual information. Nevertheless, a major limitation of these student-oriented approaches lies in their requirement to modify the teacher's weights, which is often impractical in real-world scenarios due to deployment or integrity constraints—for instance, when the teacher is provided as a black-box model or is already deployed in production environments where retraining is prohibited.

**KD with Input Perturbation:** Another prominent direction involves perturbing the input space, typically through adversarial or divergent examples during KD. In conventional KD pipelines, the teacher and student are trained on identical inputs, which may restrict the diversity of features revealed by the larger teacher model, thereby limiting the effectiveness of knowledge transfer. Heo et al. (2018a) argues that adversarial examples, crafted to align with the teacher's decision boundary, help the student learn a more accurate and generalizable boundary. However, their method generates only a single adversarial example per input, limiting its ability to explore the full spectrum of perturbations. Nguyen-Duc et al. (2023) address this limitation by formulating a teacher adversarial local distribution, which more thoroughly explores the teacher's decision boundaries, denoted as TALD. In the context of online co-distillation, Zhang et al. (2021) leverages Generative Adversarial Networks (GANs) to produce divergent examples enriched with 'dark knowledge,' thereby facilitating more effective mutual learning among co-distillation classifiers.

In contrast to these computationally intensive approaches that require access to both the original and generated datasets during distillation, Salamah et al. (2025a) proposes a lightweight alternative called CKD, which only uses a compressed version of the original dataset. CKD introduces an adaptive JPEG compression layer before the teacher model to generate multiple compressed variants of each input image using different JPEG quality factors (QFs), selecting the most informative one based on a predefined criterion. Although CKD has shown promising results, it remains limited to a small set of predefined quantization tables in a discrete space that is not optimal, thereby restricting its ability to fully explore the space of compression-induced input perturbations.

Our proposed method, **DJIP**, further develops the CKD framework to address the three key limitations discussed above:

**No Teacher Weight Updates:** DJIP further advances the coded teacher framework introduced in CKD, thus obviating the need for optimizing the teacher model's weights. This architectural decision renders DJIP particularly suitable for scenarios in which modifying the teacher is impractical, as commonly encountered in student-oriented KD paradigms.

**Greater Flexibility with Minimal Complexity:** By incorporating a differentiable JPEG layer, DJIP enables full exploration of the continuous JPEG compression space. The trained JPEG layer functions as a lightweight, modular component that can be easily attached to or removed from the model. This introduces only a small number of additional parameters, making DJIP highly deployable. In contrast to CKD, which requires repeated image-wise compression and selection across multiple QFs, DJIP performs a one-time image-wise compression without any selection process, thereby achieving significantly lower computational overhead while maintaining comparable functionality.

**Student-Oriented Design with Dynamic Optimization:** DJIP adopts the student-oriented MCMI estimator from Ye et al. (2024) and addresses its major limitation by introducing a novel alternating optimization algorithm. During training, both the class-wise clusters in the output probability space and their centroids are updated dynamically. In contrast, the fixed-centroid strategy in Ye et al. (2024), adopted as a compromise for computational tractability, leads to the accumulation of CMI estimation errors. This was evident in their experimental setup, where training was initialized from a pretrained model and conducted for only a limited number of epochs.[3] Our proposed alternating algorithm provides an analytical solution to the MCMI objective, enabling efficient centroid updates and accurate CMI estimation. Consequently, DJIP achieves performance comparable to that of MCMI, despite using significantly fewer trainable parameters.

Beyond the vision community, recent studies in large language model (LLM) distillation have revisited the foundations of KD from a divergence-optimization perspective. While developed for auto-regressive language models, these works provide insights that are broadly relevant to KD research. Wen et al. (2023) propose optimizing a general f-divergence for sequence-level distillation, highlighting the importance of selecting an appropriate divergence tailored to the structure of teacher–student discrepancies. Ko et al. (2024), Gu et al. (2023), and related efforts streamline LLM distillation pipelines by identifying training configurations that improve stability and efficiency, demonstrating that distillation effectiveness is sensitive to the choice of objective functions and optimization heuristics. More recently, Wu et al. (2025) analyzes the limitations of KL divergence in the context of heavy-tailed token distributions, revealing that KL may misallocate probability mass during distillation; Wang et al. (2025) further generalizes this idea by employing alpha-beta divergences to reshape probability allocation. Additionally, He et al. (2025) introduces difficulty-aware distillation, showing that adaptively weighting samples based on their learning difficulty can improve student performance. Although these methods are designed for LLMs, the underlying principles—such as flexible divergence design, probability-mass reallocation, and difficulty-aware weighting—offer conceptual guidance for advancing KD in computer vision settings as well.

---

[3] If the teacher model had been trained from scratch, divergence would have occurred.

## A.2 PROOF OF THEOREM 1

Recall the Markov chain $Y \to X \to \tilde{X}_w \to \hat{Y}$. To prove Theorem 1, we first introduce an auxiliary distribution $Q(\cdot \mid i, y)$ and derive a new expression for $I(\tilde{X}_w; \hat{Y} \mid Y)$ as follows:

$$
\begin{aligned}
I(\tilde{X}_w; \hat{Y} \mid Y) = I(X; \hat{Y} \mid Y) &= \sum_{y \in [C]} P_Y(y) I(X; \hat{Y} \mid Y = y) \\
&= \sum_{y \in [C]} P_Y(y) \sum_x P_{X|Y}(x \mid y) \left[ \sum_{i=1}^C P_{\hat{Y}|XY}(\hat{Y} = i \mid x, y) \times \ln \frac{P_{\hat{Y}|XY}(\hat{Y} = i \mid x, y)}{P_{\hat{Y}|Y}(\hat{Y} = i \mid Y = y)} \right] \\
&= \sum_y \sum_x P(x, y) \sum_{i=1}^C f(\tilde{x}_w)[i] \ln \frac{f(\tilde{x}_w)[i]}{P_{\hat{Y}|Y}(\hat{Y} = i \mid Y = y)} \\
&= \sum_y P_Y(y) \sum_x P_{X|Y}(x \mid y) \sum_{i=1}^C f(\tilde{x}_w)[i] \ln \frac{P_{X|Y}(x \mid y) f(\tilde{x}_w)[i]}{P_{X|Y}(x \mid y) P_{\hat{Y}|Y}(i \mid y)} \\
&= \sum_y P_Y(y) \sum_{i=1}^C P_{\hat{Y}|Y}(i \mid y) \sum_x \frac{P_{X|Y}(x \mid y) f(\tilde{x}_w)[i]}{P_{\hat{Y}|Y}(i \mid y)} \ln \frac{P_{X|Y}(x \mid y) f(\tilde{x}_w)[i]}{P_{X|Y}(x \mid y) P_{\hat{Y}|Y}(i \mid y)} \quad (11) \\
&= \max_{\{Q(\cdot|i,y)\}} \sum_y P_Y(y) \sum_{i=1}^C P_{\hat{Y}|Y}(i \mid y) \sum_x \frac{P_{X|Y}(x \mid y) f(\tilde{x}_w)[i]}{P_{\hat{Y}|Y}(i \mid y)} \ln \frac{Q(x \mid i, y)}{P_{X|Y}(x \mid y)} \\
&= \max_{\{Q(\cdot|i,y)\}} \sum_{x,y} P(x, y) \sum_{i=1}^C f(\tilde{x}_w)[i] \ln \frac{Q(x \mid i, y)}{P_{X|Y}(x \mid y)}, \quad (12)
\end{aligned}
$$

where Equation 11 follows from the cross entropy inequality, and the maximization in Equation 12 is achieved when

$$
Q^*(x \mid i, y) = \frac{P_{X|Y}(x \mid y) f(\tilde{x}_w)[i]}{P_{\hat{Y}|Y}(i \mid y)}. \quad (13)
$$

Thus, the single minimization problem in equation 5 can be converted into a double minimization problem over $w$ and $\{Q(\cdot \mid i, y)\}$ as follows:

$$
\begin{aligned}
&\min_w \left\{ \mathbb{E}_X H(P_{Y|X}, f(\tilde{X}_w)) - \lambda\, I(\tilde{X}_w; \hat{Y} \mid Y) \right\} \\
&= \min_w \left\{ \mathbb{E}_X H(P_{Y|X}, f(\tilde{X}_w)) - \lambda\, I(X; \hat{Y} \mid Y) \right\} \\
&= \min_w \left\{ \mathbb{E}_X H(P_{Y|X}, f(\tilde{X}_w)) - \lambda \max_{\{Q(\cdot|i,y)\}} \sum_{x,y} P(x, y) \sum_{i=1}^C f(\tilde{x}_w)[i] \ln \frac{Q(x \mid i, y)}{P_{X|Y}(x \mid y)} \right\} \\
&\equiv \min_w \left\{ \mathbb{E}_X H(P_{Y|X}, f(\tilde{X}_w)) - \lambda \max_{\{Q(\cdot|i,y)\}} \sum_{x,y} P(x, y) \sum_{i=1}^C f(\tilde{x}_w)[i] \ln Q(x \mid i, y) \right\} \quad (14) \\
&= \min_w \left\{ \mathbb{E}_X H(P_{Y|X}, f(\tilde{X}_w)) + \lambda \min_{\{Q(\cdot|i,y)\}} -\sum_{x,y} P(x, y) \sum_{i=1}^C f(\tilde{x}_w)[i] \ln Q(x \mid i, y) \right\} \\
&= \min_w \min_{\{Q(\cdot|i,y)\}} \left\{ \sum_x P(x) H(P_{Y|X}(\cdot \mid x), f(\tilde{x}_w)) - \lambda \sum_{x,y} P(x, y) \sum_{i=1}^C f(\tilde{x}_w)[i] \ln Q(x \mid i, y) \right\}, \quad (15)
\end{aligned}
$$

where the symbol "$\equiv$" in Equation 14 indicates equivalence up to an additive constant independent of $Q(x \mid i, y)$ and $w$, which is omitted here for clarity.

## A.3 PSEUDO-CODE FOR ALTERNATING ALGORITHM

The pseudo-code of the alternating optimization algorithm is provided in Algorithm 1. In Section 6.1, we present comprehensive experiments on CIFAR-100 that demonstrate its advantages over the fixing-centroid method in Ye et al. (2024) and validate the effectiveness of incorporating a differentiable JPEG layer.

---

**Algorithm 1** The proposed alternating algorithm for solving the optimization problem in Theorem 1.

---

**Input:** DNN $f_\theta$ as $f$; JPEG layer $\mathcal{J}_d$ with trainable parameters $w$; training set $\mathcal{D} = \{(x_i, y_i)\}_{i=1}^N$ with its $B$ mini-batches $\{\mathcal{B}^b\}_{b \in [B]}$; class labels $C$; number of epochs $T$ and hyper-parameter $\lambda$.

1: Initialize $w^0$.
2: **repeat**
3:     **for** $b = 1$ to $B$ **do**
4:         [*Update S*]:
5:         Fix $w^{b-1}$. Update centroids $\{S_y^b\}_{y \in [C]}$ according to Equation 9:

$$S_y^b[i] \leftarrow \frac{1}{|\mathcal{D}_y|} \sum_{x_j \in \mathcal{D}_y} f(\mathcal{J}_d(x_j, w^{b-1}))[i], \quad \forall i, y \in [C]. \tag{16}$$

6:         [*Update Q*]:
7:         Fix $w^{b-1}$. Calculate $Q^b(x \mid i, y)$, according to Equation 10:

$$Q^b(x \mid i, y) \leftarrow \frac{\frac{|\mathcal{D}_{x,y}|}{|\mathcal{D}_y|} f(\mathcal{J}_d(x, w^{b-1}))[i]}{S_y^b[i]}, \quad \forall (x, y) \in \mathcal{B}^b, i \in [C]. \tag{17}$$

8:         [*Update w*]
9:         Fix $\{Q^b(\cdot \mid i, y)\}_{(x,y) \in \mathcal{B}^b}^{i \in [C]}$. Update weights $w^{b-1}$ to $w^b$ by using SGD over the objective
    function

$$\mathcal{L}_{\mathcal{B}^b}(\lambda, w^{b-1}, \{Q^b(\cdot \mid i, y)\}_{(x,y) \in \mathcal{B}^b}^{i \in [C]}).$$

10:     **end for**
11:     Set $w^0 \leftarrow w^B$
12: **until** $T$ epochs are completed

13: **return** Trained parameters $w^B$.

---

## A.4   Convergence Analysis and Empirical Evidence

As discussed above, the inner minimization step in Equation 15 satisfies the cross-entropy inequality. Suppose the impact of the random mini-batch sampling and SGD is ignored. In that case, the alternating algorithm is guaranteed to converge in theory, since given $w$, the optimal $Q^*(x \mid i, y)$ can be found analytically via 13. Although in practice the alternating algorithm may not converge to a global minimum, this is not the limitation of our algorithm, but the nature of all SGD-based deep learning algorithms.

Empirically, our alternating procedure exhibits stable behavior throughout training. As shown in Figure 3, the training CMI value consistently improves, and the training DJIP loss and the overall training loss converge smoothly over iterations on both ResNet-34 and ResNet-152. These observations indicate that the proposed alternating algorithm behaves stably in practice at scale.

## A.5   Implementation Details

### A.5.1   Setups for JPEG Layer Training Stage

We follow the implementation and design of the differentiable JPEG layer from Salamah et al. (2025b), but set the minimum quantization step to 1, which is more commonly used in reality, instead of 0. The stochastic gradient descent (SGD) optimizer is used with a momentum of 0.9 and a weight decay of $5 \times 10^{-4}$ for all JPEG layer training experiments.

A key feature of the differentiable soft quantizer $\mathcal{Q}_d$ in the JPEG layer is its variable softness, controlled by the trainable sharpness parameter $\alpha$. However, as shown in Salamah et al. (2025b), when $\alpha$ is sufficiently large, its gradient vanishes, preventing effective updates. Therefore, we initialize all entries of $\boldsymbol{\alpha}$ to a large constant value of 20 and exclude $\boldsymbol{\alpha}$ from training. Thus, the trainable parameters $w$ of the DJIP teacher are the quantization tables $\boldsymbol{Q}$, which we initialize with all ones.

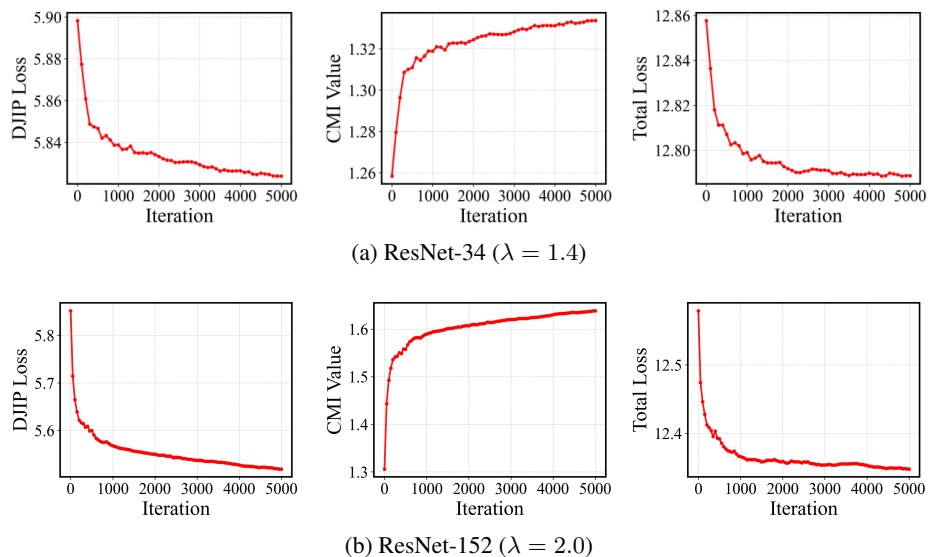

(a) ResNet-34 ($\lambda = 1.4$)

(b) ResNet-152 ($\lambda = 2.0$)

Figure 3: Convergence curves of the proposed alternating optimization on ImageNet. The training DJIP loss, training CMI value, and total training loss exhibit stable and monotonic trends, confirming the empirical convergence of our method.

For **CIFAR-100**, all pretrained teachers are adopted from the official repository of Tian et al. (2019). We use a learning rate of 0.1 and train for 20 epochs with a batch size of 32 over 2 GPUs (total batch size 64). For all CIFAR-100 experiments comparing with the CE teacher, including those involving ViT models, we set the hyperparameter $\lambda$ in the objective function 8 to 0.5 by default. An exception is made for the VGG-13 and MobileNetV2 teacher-student pair in Table 2, where we set $\lambda = 0.6$. To demonstrate that DJIP is further orthogonal to MCMI in Table 6, we train the DJIP-MCMI teacher with $\lambda = 0.3$. Throughout the alternating training process, we always use the full training set $\mathcal{D}$ to update the centroids as proposed in Algorithm 1 with the update interval $= 5$.

For **ImageNet**, we adopt pretrained teachers from the PyTorch official repository (Paszke et al., 2019). DJIP teachers are trained for one epoch (5005 iterations) with a learning rate of 0.01, using 4 GPUs and a batch size of 64 per GPU (total batch size 256). We experiment with various values of $\lambda$, namely $\{0.4, 1.0, 1.4, 2.0, 2.4, 3.0\}$, and select $\lambda = 1.4$ for ResNet-34 and $\lambda = 2.0$ for ResNet-50.

When applied to large-scale datasets such as ImageNet, updating centroids at each iteration (update interval $= 1$.) via Algorithm 1 becomes computationally expensive. To reduce this complexity, we construct a balanced sub-dataset $\hat{\mathcal{D}}$ by randomly sampling $|\hat{\mathcal{D}}_y| = 16$ instances per class from the training set. By replacing the full training set in Equation 9 with $\hat{\mathcal{D}}$, a lightweight variant of the alternating algorithm can be derived. In the first iteration, the dummy distribution $\{Q(\cdot \mid i, y)\}$ is updated using the entire training set. Thereafter, it is updated using the sub-dataset and an exponential moving average (EMA) with a smoothing factor $\alpha = 0.9$.

Thus, the resulting pseudo-code on ImageNet experiments is similar to Algorithm 1, except that Equation 16 is replaced by:

$$S_y^b[i] \leftarrow \alpha \times S_y^{b-1}[i] + (1 - \alpha) \times \frac{1}{|\hat{\mathcal{D}}_y|} \sum_{x_j \in \hat{\mathcal{D}}_y} f(\mathcal{J}_d(x_j, w^{b-1}))[i], \quad \forall i, y \in [C]. \tag{18}$$

To further handle the increased complexity of ImageNet tasks compared to CIFAR-100 tasks, following the setups of Salamah et al. (2025b), we incorporate five rounds of $\mathcal{Q}_d$ quantization operation with independent trainable parameters inside the differentiable JPEG layer to expand the compression search space. Moreover, we follow the gradient magnitude control method proposed by Salamah et al. (2025b), with the Gradient Scaling Constants $\hbar_m$ set to 20 to control the gradient magnitude to ensure more stable updates for $Q$.

### A.5.2 SETUPS FOR STUDENT DISTILLATION STAGE

For experimental setups of all knowledge distillation variants in this paper, we use SGD with momentum 0.9 and weight decay of $1 \times 10^{-4}$ as the optimizer.

For **CIFAR-100**, we follow the distillation setups from Tian et al. (2019). Specifically, we train the student models for 240 epochs with a batch size of 64 on 1 GPU. The initial learning rate is set to 0.05 and decayed by a factor of 0.1 at epochs 150, 180, and 210. For MobileNetV2, ShuffleNetV1, and ShuffleNetV2, we use a smaller initial learning rate of 0.01.

For **ImageNet**, we adopt the training configuration from Zhao et al. (2022): 100 epochs, a learning rate of 0.1, and a batch size of 256 on a single GPU. The only exception is ReviewKD (Chen et al., 2021), which uses a batch size of 128 and is trained on two GPUs.

For both datasets, we evaluated several state-of-the-art distillation methods using their official implementations and reported hyper-parameters, and successfully reproduced the results reported in their original papers.

### A.6 ADDITIONAL ABLATION STUDIES ON DJIP HYPER-PARAMETERS

In this section, we conduct a series of ablation studies to investigate the sensitivity of hyper-parameters and how those key hyper-parameters affect the behavior of the DJIP teacher and the resulting student performance. Specifically, we examine the influence of the weighting coefficient $\lambda$, the number of quantization rounds in JPEG, the learning rate for JPEG layer's updates, and the cluster centroid update interval.

### A.6.1 EFFECT OF $\lambda$ IN DJIP

In Table 7, we vary $\lambda$ in $\{0.4, 1.0, 1.4, 2.0, 2.4, 3.0\}$ to evaluate how the balance between CMI maximization and CE minimization affects teacher behavior. Larger values of $\lambda$ increase the teacher's CMI while also raising its CE loss. The student's Top-1 accuracy follows a quasiconcave trend and peaks at $\lambda = 1.4$, indicating that this value provides an effective trade-off between informativeness and predictive correctness.

Table 7: Effect of $\lambda$ on student test accuracy (%), the CMI and CE losses of the DJIP teacher. The results are compared with those of a standard CE teacher in the ResNet-34 $\rightarrow$ ResNet-18 setting on ImageNet with vanilla KD. **Bold** numbers indicate the best performance.

| KD | CE | DJIP with $\lambda$ equals to | | | | | | |
|---|---|---|---|---|---|---|---|---|
| | | 0.4 | 1.0 | 1.4 | 2.0 | 2.4 | 3.0 | 4.0 |
| Teacher CELoss | 0.560 | 0.568 | 0.568 | 0.582 | 0.630 | 0.595 | 0.627 | 0.871 |
| Teacher CMI | 0.7180 | 0.7257 | 0.7268 | 0.7382 | 0.7869 | 0.7523 | 0.7825 | 0.9866 |
| Student Acc | 70.660 | 71.362 | 71.452 | **71.654** | 71.274 | 71.370 | 71.290 | 70.694 |

### A.6.2 EFFECT OF NUMBER OF ROUNDS IN DJIP

We investigate the effect of increasing the number of quantization rounds in JPEG. Here, one round refers to applying quantization to the DCT coefficients once using a separate quantization table. Introducing multiple rounds effectively expands the perturbation space, which is particularly beneficial for complex datasets such as ImageNet. As the number of rounds increases, the teacher's CMI consistently improves; however, excessive rounds can lead to elevated CE loss, indicating a loss of predictive reliability. As shown in Table 8, the student's accuracy grows steadily from 1 to 5 rounds and reaches its peak at 5 rounds, suggesting that a moderate degree of iterative quantization provides the most favorable balance between informativeness and stability.

### A.6.3 EFFECT OF LEARNING RATE IN DJIP

We evaluate teacher learning rates of 0.1, 0.02, 0.01, and 0.001. As shown in Table 9, an overly high learning rate leads to unstable JPEG layer updates, manifested by excessively large CMI values and noticeably increased CE loss, which ultimately undermines the student's performance. The learning

Table 8: Effect of the number of quantization rounds on student accuracy (%) and on the CMI and CE losses of the DJIP teacher. Results are reported for the ResNet-34 $\rightarrow$ ResNet-18 setting on ImageNet under vanilla KD. **Bold** numbers denote the best performance.

| KD | CE | DJIP with number of rounds equals to | | | |
|---|---|---|---|---|---|
| | | 1 | 3 | 5 | 6 |
| Teacher CELoss | 0.560 | 0.560 | 0.561 | 0.582 | 0.768 |
| Teacher CMI | 0.7180 | 0.7178 | 0.7185 | 0.7382 | 0.9097 |
| Student Acc | 70.660 | 71.278 | 71.610 | **71.654** | 70.960 |

rate of 0.01 yields the best overall results, indicating that a stable yet sufficiently responsive update regime is essential for effective distillation.

Table 9: Effect of learning rate on student accuracy (%), along with the CMI and CE losses of the DJIP teacher. Results are reported for the ResNet-34 $\rightarrow$ ResNet-18 setting on ImageNet under vanilla KD. **Bold** indicates the best performance.

| KD | CE | DJIP with learning rate equals to | | | |
|---|---|---|---|---|---|
| | | 0.1 | 0.02 | 0.01 | 0.001 |
| Teacher CELoss | 0.560 | 0.875 | 0.601 | 0.582 | 0.561 |
| Teacher CMI | 0.7180 | 0.9961 | 0.7577 | 0.7382 | 0.7191 |
| Student Acc | 70.660 | 70.628 | 71.444 | **71.654** | 71.432 |

#### A.6.4 EFFECT OF CENTROID UPDATE INTERVAL IN DJIP

We further study the impact of the centroid update interval of DJIP in Table 10. When using intervals of 3, 5, 50, and 500 iterations, we observe that the student accuracy remains largely stable, with the best results appearing at intervals of 3 and 5. Although both intervals yield similar accuracy, we select the interval of 5 due to efficiency concerns. As the update interval becomes longer, the student's accuracy shows a mild decrease; however, it consistently surpasses that obtained with the standard CE teacher. This suggests that moderately frequent centroid updates provide more reliable centroid estimates while avoiding unnecessary computation, thereby enabling a more accurate inner minimization of Equation 15 and ultimately improving distillation performance.

Table 10: Effect of centroid update interval on student accuracy (%) on CIFAR-100 (averaged over 3 runs). Results are reported for the ResNet-34 $\rightarrow$ ResNet-18 setting, with update intervals of 3, 5, 50, and 500 iterations. All configurations consistently outperform the standard CE teacher, with the best performance achieved at an interval of 3 and 5.

| Teacher | Student | Method | CMI | KD | DKD | DIST | CC | RKD | AT | FitNet | FT | SP | ITRD | CRD |
|---|---|---|---|---|---|---|---|---|---|---|---|---|---|---|
| | | CE | 0.009 | 73.81 | 73.94 | 74.11 | 70.25 | 71.50 | 71.84 | 69.39 | 70.29 | 73.34 | 75.49 | 74.30 |
| | | Interval=3 | 0.341 | 74.49 | 75.87 | 74.58 | 70.92 | 71.87 | 72.50 | 69.50 | 71.29 | 73.58 | 76.02 | 74.56 |
| ResNet-50 | VGG-8 | Interval=5 | 0.341 | 74.48 | 75.87 | 74.80 | 70.92 | 71.93 | 72.42 | 69.57 | 71.24 | 73.87 | 75.91 | 74.55 |
| | | Interval=50 | 0.341 | 74.33 | 75.58 | 74.63 | 70.78 | 71.81 | 72.38 | 69.35 | 71.23 | 73.68 | 75.91 | 74.24 |
| | | Interval=500 | 0.342 | 74.32 | 75.49 | 74.52 | 70.68 | 71.73 | 72.37 | 69.02 | 71.06 | 73.61 | 75.81 | 74.21 |

#### A.7 DJIP WITH OTHER INPUT PERTURBATION METHOD

As stated in Theorem 1, the parameter $\omega$ represents arbitrary perturbation parameters and is fully agnostic to the specific choice of input perturbation mechanism. Consequently, the proposed alternating optimization framework is not restricted to JPEG quantization tables. The differentiable JPEG module adopted in our experiments serves merely as a convenient instantiation, selected due to its simplicity, computational efficiency, and the availability of robust differentiable implementations. In principle, this perturbation layer may be replaced by any differentiable codec. As another illustrative example, we additionally incorporate a convolutional autoencoder as the input perturbation module.

We employ a symmetric convolutional autoencoder designed for CIFAR-resolution images. The encoder is composed of convolution, batch normalization, and ReLU layers, followed by a max-

pooling operation in the middle. The decoder mirrors this structure using convolution and transposed-convolution layers to upsample feature maps and recover the original spatial resolution. As observed in Section A.10, the optimized quantization tables consistently exhibit small step sizes. Motivated by this, we choose a latent dimensionality comparable to the input, enabling the autoencoder to learn an identity-preserving or denoising transformation. A final sigmoid activation constrains the reconstructed output to the normalized pixel range.

For pretraining, the autoencoder is trained on CIFAR-100 with standard data augmentation. Optimization is performed using Adam with a learning rate of $1 \times 10^{-3}$, and the reconstruction objective is mean squared error (MSE). Training proceeds for 200 epochs with a batch size of 64. On the CIFAR-100 test set, the pretrained autoencoder achieves a PSNR of $41.52\,\mathrm{dB}$, an SSIM of $0.9961$, and an MSE of $8.3 \times 10^{-5}$, indicating high-fidelity reconstruction.

Following the training framework in Figure 1, we insert the autoencoder before the classifiers and optimize its parameters using the alternating optimization algorithm. We train for 20 epochs with a learning rate of $1 \times 10^{-3}$ and set $\lambda = 0.3$. After optimizing the perturbation module, standard distillation is performed. The final results are reported in Table 11.

Table 11: The test accuracy (%) of students on CIFAR-100 with autoencoder as input perturbation method.

| Teacher | Student | Method | CMI | KD | FitNet | FT | SP | CRD | CC | RKD |
|---------|---------|--------|-----|-----|--------|-----|-----|-----|-----|-----|
| ResNet-34 | ResNet-18 | CE | 0.015 | 72.98 | 71.02 | 70.58 | 72.68 | 73.94 | 70.71 | 71.48 |
| 74.64 | 70.36 | DJIP | 0.067 | 73.31 | 71.31 | 71.03 | 72.92 | 74.05 | 71.13 | 71.77 |
|  |  | $\triangle$ | $\nearrow$ | +0.33 | +0.29 | +0.45 | +0.24 | +0.11 | +0.42 | +0.29 |

## A.8 DJIP ONLINE DISTILLATION COMPLEXITY

As mentioned in Figure 1, DJIP contains 2 stages. The first stage trains the JPEG layer in an offline manner. Since the learned JPEG layer can be reused for training multiple student models, this stage is executed only once and its training time is amortized and therefore negligible in the overall framework.

For the second online stage, we present the student training throughput, GPU peak memory, and total runtime in Table 12. The student distillation stage is conducted on a single RTX 2080 Ti GPU.

Table 12: Comparison of DJIP distillation complexity measured with different teacher-student pairs on CIFAR-100 with vanilla KD method and TALD, following the configuration specified in Section A.5.

| Teacher Student | Method | Throughput (img/ms) | Peak GPU Memory (MB/GPU) | Total Runtime (s) |
|-----------------|--------|---------------------|--------------------------|-------------------|
| VGG-13 | Vanilla | 7.11 | 614 | 2026.6 |
|  | DJIP | 5.60 | 624 | 2572.8 |
| VGG-8 | TALD | 2.48 | 1060 | 5829.6 |
| ResNet-56 | Vanilla | 4.61 | 408 | 3124.8 |
|  | DJIP | 3.70 | 410 | 3895.2 |
| ResNet-20 | TALD | 1.38 | 946 | 10447.2 |
| WRN-40-2 | Vanilla | 3.53 | 668 | 4084.8 |
|  | DJIP | 2.98 | 670 | 4826.4 |
| WRN-40-1 | TALD | 1.15 | 1262 | 12516.0 |

## A.9 FEW SHOT CLASSIFICATION

In conventional few-shot classification, only a $\beta$ percent of instances from each class are made available for model training (Luo et al., 2023). Translating this idea into the KD setting, the JPEG layer is trained on the full dataset, while only a $\beta$ percent subset of samples per class is used to train the student during distillation.

To examine the effectiveness of the proposed DJIP teacher under limited-data conditions, we conduct experiments on CIFAR-100 using WRN-40-2 WRN-16-2 teacher-student pair. We evaluate several values of $\beta$, namely $5, 10, 20, 50, 75$, over the vanilla KD method. The results are presented in Table 13. As shown, the student consistently benefits from distillation with the DJIP teacher, and the improvement is especially notable in more challenging low-data regimes (smaller $\beta$). These results further support the robustness of our method under constrained data scenarios.

Table 13: Comparison of student accuracy (%) under few-shot distillation settings on CIFAR-100. The teacher is trained on the full dataset, while only a $\beta$ percent subset of samples per class is used to train the student.

| Teacher | Student | Method | $\beta$ | | | | |
|---|---|---|---|---|---|---|---|
| | | | 5 | 10 | 20 | 50 | 75 |
| WRN-40-2 | WRN-16-2 | CE | 34.75 | 50.73 | 60.93 | 69.60 | 72.17 |
| | | DJIP | 40.56 | 53.95 | 62.98 | 70.87 | 73.15 |
| | | $\Delta$ | +5.81 | +3.22 | +2.06 | +1.27 | +0.98 |

## A.10 TRAINED QUANTIZATION TABLE VISUALIZATION

The quantization tables trained for the DJIP teacher, corresponding to the results in Tables 1, 2, and 3, are presented in Figures 4 and 5. These results indicate that, across different datasets, the models consistently apply stronger compression to the Y channel than to the Cb and Cr channels, which in turn leads to higher CMI values after training.

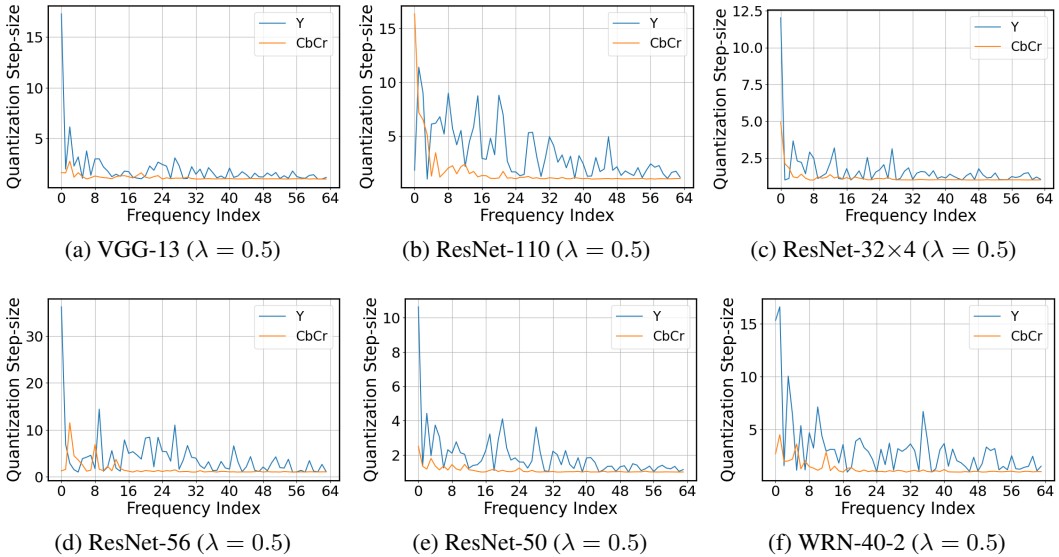

(a) VGG-13 ($\lambda = 0.5$)    (b) ResNet-110 ($\lambda = 0.5$)    (c) ResNet-32×4 ($\lambda = 0.5$)

(d) ResNet-56 ($\lambda = 0.5$)    (e) ResNet-50 ($\lambda = 0.5$)    (f) WRN-40-2 ($\lambda = 0.5$)

Figure 4: Trained quantization tables on CIFAR-100 models used in Table 1 and 2.

## A.11 VISUALIZE THE OUTPUT PROBABILITY SPACE

We visualize how the output probability distributions of the ResNet-32×4 DJIP teacher evolve across epochs during a single training run, as illustrated in Figure 6. The simplex is visualized by projecting the 3-class softmax probability vectors onto a 2D plane, where the one-hot vectors of each class are mapped to the vertices of an equilateral triangle. Temperature scaling with $T = 4.0$ is applied to the softmax outputs.

Initially, the teacher produces highly confident predictions resembling one-hot vectors, with output probabilities concentrated near the corners of the simplex. As training progresses, facilitated by the differentiable JPEG layers, both the output distributions and the centroids of each class cluster

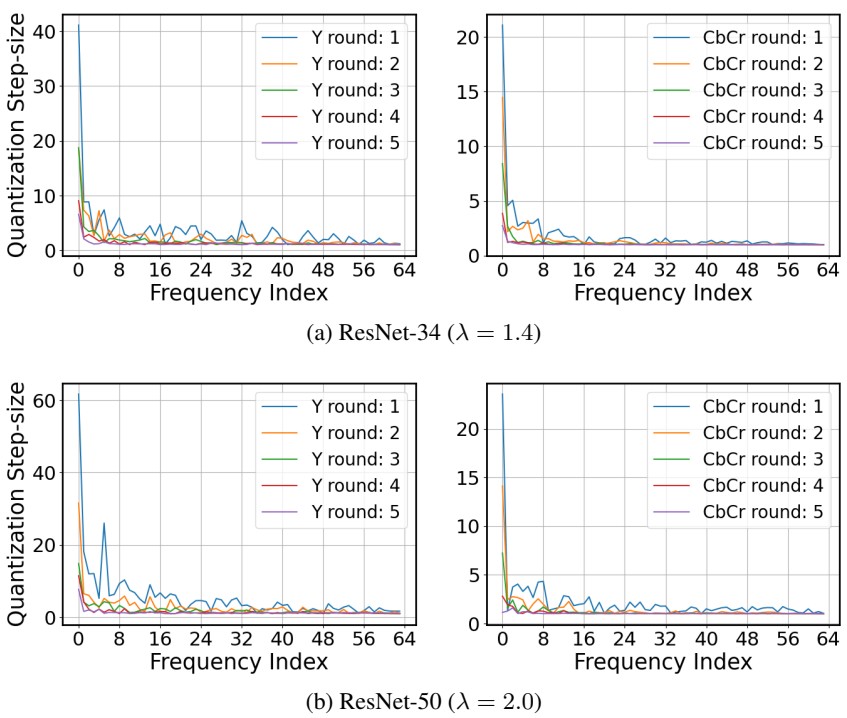

(a) ResNet-34 ($\lambda = 1.4$)

(b) ResNet-50 ($\lambda = 2.0$)

Figure 5: Trained quantization tables on ImageNet models in Table 3.

gradually shift toward the center of the simplex, leading to an increase in CMI values. While maintaining correct classifications, the predictions become less confident yet more informative. Figure 6 further illustrates the convergence of centroid trajectories, thereby validating the convergence of the alternating optimization algorithm.

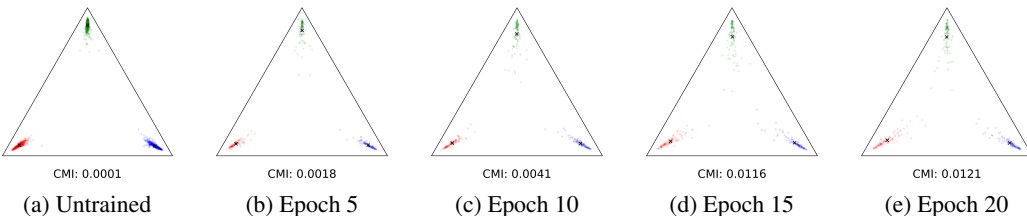

(a) Untrained     (b) Epoch 5     (c) Epoch 10     (d) Epoch 15     (e) Epoch 20

Figure 6: The output probability vectors of the trained DJIP teacher for a three-class classification task are visualized on a 2-simplex over different epochs within a single training run. Three classes are randomly selected from the CIFAR-100 training set, with 100 samples per class. Temperature scaling with $T = 4.0$ is applied to the softmax outputs to enhance distributional smoothness. In the plots, cross markers denote the class centroids, and the CMI values are computed based solely on these three selected classes.

A more comprehensive comparison of the output probability space between the CE teacher and the DJIP teacher is shown in Figure 7, with temperature scaling $T = 4.0$ as well. We observe that the clusters corresponding to the DJIP teacher become less concentrated.

## A.12 THE USE OF LARGE LANGUAGE MODELS (LLMS)

In this work, we only employed LLMs to assist with writing polish and refinement, as well as for literature retrieval and discovery (e.g., identifying related work).

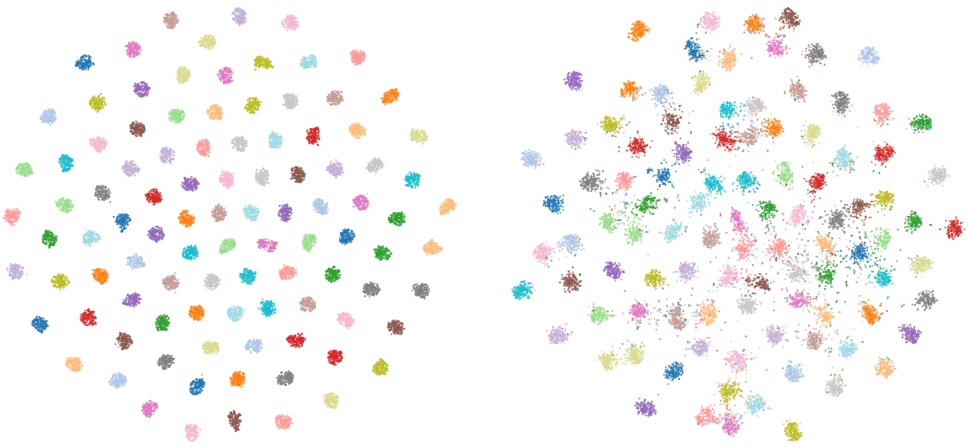

(a) ResNet-32x4 (Left: CE teacher; Right: DJIP teacher)

Figure 7: t-SNE (van der Maaten & Hinton, 2008) visualization of features extracted from the CIFAR-100 training set with 500 samples per cluster in all 100 categories.

