# OpenReview forum: "Differentiable JPEG-based Input Perturbation for Knowledge Distillation Amplification via Conditional Mutual Information Maximization"
_ICLR.cc/2026/Conference — ICLR 2026 Poster_

### Official Review · Reviewer_gUa4 · 2025-10-31

**Soundness:** 3
**Presentation:** 3
**Contribution:** 2
**Rating:** 6
**Confidence:** 3

**Summary:**

This paper presents Differentiable JPEG-based Input Perturbation (DJIP), a plug-and-play framework that boosts knowledge distillation by maximizing a frozen teacher’s conditional mutual information (CMI) without altering its weights. DJIP inserts a trainable, differentiable JPEG module before the teacher to learn input-space perturbations, and pairs it with an alternating optimization routine that dynamically updates class centroids to maximize perturbed CMI. On CIFAR-100 and ImageNet—spanning CNNs and ViTs and multiple distillers (KD, DKD, AT)—DJIP delivers consistent student accuracy improvements (up to 4.11%), while composing cleanly with existing methods.

Contributions
1. A lightweight, broadly compatible input-perturbation mechanism via differentiable JPEG compression.
2. An alternating CMI-maximization algorithm that overcomes fixed-centroid limitations (e.g., MCMI).
3. Extensive empirical validation demonstrating orthogonality to prevailing techniques and scalability across datasets, architectures, and distillers.

**Strengths:**

1. Originality. DJIP reframes JPEG-based input perturbation as a continuous optimization problem, in contrast to prior discrete toggling or adversarial noise schemes. The proposed alternating routine offers a principled advance for CMI maximization/estimation, jointly adapting perturbations and class centroids to better capture teacher–input dependence.
2. Quality. The evaluation is broad and careful: two large-scale datasets, 10+ architectures (CNNs and ViTs), and 15+ KD methods, with component-wise ablations (Table 5) that isolate the contribution of the differentiable JPEG layer and the centroid updates. A thorough hyperparameter sweep (Appendix A.5) tests sensitivity and demonstrates stable gains across settings, strengthening the empirical claims.
3. Clarity. Exposition is accessible: the CMI objective is introduced with clear intuition and notation (Sec. 3.2), and the overall pipeline is modularized in Fig. 1, making it straightforward to slot DJIP before a frozen teacher. Algorithm boxes and training pseudocode further improve reproducibility.
4. Significance. Because DJIP is plug-and-play and lightweight (≈5–12% parameter overhead), it is practical for resource-constrained deployments. Notably, it narrows or even surpasses larger baselines (e.g., a 3B DJIP student outperforming a 14B non-DJIP counterpart), and its benefits persist across diverse backbones and distillers—evidence of orthogonality rather than method-specific tuning. Collectively, these attributes position DJIP as a scalable, implementation-friendly upgrade path for KD pipelines.

**Weaknesses:**

1. Computational Efficiency. Table 4 reports parameter overhead, but end-to-end latency, peak memory, and throughput under realistic batch sizes/hardware are not measured. These metrics are essential for edge/smartphone deployment. Suggest reporting wall-clock inference (ms/img), GPU/CPU memory footprints, and A/B latency deltas with and without DJIP across batch sizes.
2. Baseline Diversity. Comparisons emphasize CKD and MCMI, but omit recent non-JPEG perturbation approaches (e.g., GAN/ diffusion-generated counterfactuals, learned augmentors). Including such baselines would clarify whether DJIP’s gains stem from JPEG structure or from perturbative training per se.
3. Theoretical Limits. Convergence evidence (Fig. 5) is empirical; there is no formal rate or stationarity guarantee for the alternating updates. A brief analysis (e.g., monotonic ascent under bounded curvature, or conditions ensuring convergence to a critical point) would solidify the algorithmic contribution.
4. Data Efficiency. Experiments rely on high-quality data (e.g., PixMo-Cap); low-data regimes (few-shot/long-tail KD) remain untested. Evaluating DJIP under subsampling, noisy labels, or class imbalance—and reporting data-efficiency curves—would demonstrate robustness where KD is most needed.

**Questions:**

1. Computational Overhead & Scaling. How does DJIP’s end-to-end cost scale with input resolution and batch size relative to vanilla KD? Please report latency (ms/img), throughput (img/s), and peak memory across GPU/CPU settings. Have you explored hardware-aware optimizations of the JPEG layer (e.g., fused CUDA kernels, SIMD/Neon intrinsics, TensorRT plugins), and what is the measured speedup?
2. Format Generalization. Beyond JPEG, have you evaluated differentiable alternatives (e.g., WebP/AVIF or learned compression) to test the mechanism’s generality? A head-to-head comparison controlling bitrate/PSNR would clarify whether gains stem from JPEG’s structure or from compression-induced perturbations more broadly.
3. Algorithmic Stability. Does the alternating scheme exhibit instability when class centroids shift quickly (e.g., early training, long-tail classes)? Would momentum/EMA, trust-region or proximal updates, or line-search on the CMI objective improve stability and convergence consistency? Any failure cases or sensitivity analyses?
4. Black-Box Teachers. How does DJIP perform when the teacher is accessible only via queries (API/closed-source), precluding gradient flow? Can zeroth-order/finite-difference estimates, score-matching proxies, or pseudo-label distillation approximate the CMI objective, and with what accuracy–cost trade-offs?

---

> ### Author Response · Authors · 2025-11-25
> **Reply to Reviewer gUa4 (part 1)**
>
> Dear Reviewer **gUa4**:
>
> We thank you for taking the time to review our paper and provide valuable feedback. Below, please find our responses to your comments.
>
>
> ## Strengths
> >S4. Significance. Because DJIP is plug-and-play and lightweight (≈5–12% parameter overhead), it is practical for resource-constrained deployments. Notably, it narrows or even surpasses larger baselines (e.g., a 3B DJIP student outperforming a 14B non-DJIP counterpart), and its benefits persist across diverse backbones and distillers—evidence of orthogonality rather than method-specific tuning. Collectively, these attributes position DJIP as a scalable, implementation-friendly upgrade path for KD pipelines.
>
> **Reply**: We highly appreciate your positive summary of the strengths of our method. We would like to make two clarifications. First, even for a relatively small student model used in our experiments (e.g., ResNet-56 on CIFAR-100 with approximately 0.86M parameters), the parameter overhead introduced by DJIP is even smaller than stated—less than **0.015%**—since we only add 128 additional parameters before the model. Second, we would like to clarify a potential misunderstanding: our experiments do not involve 3B–14B models. All experiments in the current submission are conducted on models at the **million-parameter scale**. We will revise the manuscript to avoid potential confusion.
>
>
> ## Weaknesses
> >W1.  Computational Efficiency. Table 4 reports parameter overhead, but end-to-end latency, peak memory, and throughput under realistic batch sizes/hardware are not measured. These metrics are essential for edge/smartphone deployment. Suggest reporting wall-clock inference (ms/img), GPU/CPU memory footprints, and A/B latency deltas with and without DJIP across batch sizes.
>
> **Reply**: Thank you for raising this concern. We would first like to clarify that **Table 4** does **not** report parameter overhead. Instead, it summarizes our CIFAR-100 results under different teacher–student paradigms (ViT-to-CNN, CNN-to-ViT, and ViT-to-ViT), demonstrating that DJIP remains effective across heterogeneous architectures, particularly for distilling ViTs.
>
> Regarding deployment efficiency, we emphasize that the JPEG module in DJIP functions purely as an _input perturbation layer during training_. It is **not** part of the teacher model. Once training is complete and the JPEG layer is removed, the teacher model remains identical. More importantly, we do not introduce any additional parameters or computational components into the student network itself. As a result, DJIP adds **no extra latency, memory usage, or computational overhead** at inference time, and the deployment process is unchanged from the student under vanilla KD.
>
> The only extra cost lies on the **training side**, and DJIP’s training pipeline contains two stages:
>
> As shown in Figure 1, the **first** stage trains the JPEG layer in an offline manner. Since the learned JPEG layer can be reused for training multiple student models, this stage is executed only once, so its training time is amortized and becomes negligible overall.
>
> For the **second** student distillation stage, following your suggestion, we report the student distillation throughput, GPU peak memory, and total runtime in **Table 12** of the revised manuscript. This experiment is conducted on a single RTX 2080 Ti GPU.
>
> **Table 12**: Comparison of distillation complexity on VGG-13 $\Rightarrow$ VGG-8 pairs and CIFAR-100 dataset with vanilla KD method and another perturbation method TALD.
> | KD Method  | Throughput (img/ms) | Peak GPU Memory (MB) | Total Time (s) |
> |------------|---------------------|----------------------|----------------|
> | Vanilla KD | 7.11    					   | 614                  | 2026.6         |
> | DJIP       | 5.60   				 		 | 624                  | 2572.8         |
> | TALD       | 2.48    				 	 	 | 1060                 | 5829.6         |
>
> These results show that DJIP incurs **modest** overhead during training, and its training cost is significantly lower than other perturbation-based KD methods (e.g., TALD).

---

> ### Author Response · Authors · 2025-11-25
> **Reply to Reviewer gUa4 (part 2)**
>
> >W2. Baseline Diversity. Comparisons emphasize CKD and MCMI, but omit recent non-JPEG perturbation approaches (e.g., GAN/ diffusion-generated counterfactuals, learned augmentors). Including such baselines would clarify whether DJIP’s gains stem from JPEG structure or from perturbative training per se.
>
> **Reply**: Thank you for the insightful comment. Although the paper presents DJIP using a differentiable JPEG module, the underlying framework is highly **generic** and accommodates a broad class of input perturbation mechanisms, including JPEG and autoencoder-based models. Our theoretical results—Theorem 1 and the alternating CMI maximization algorithm—hold in this general setting. The JPEG module used in our experiments is merely one concrete instantiation of this perturbation mechanism, chosen for its simplicity and computational efficiency; it can be replaced by any differentiable codec without altering the core formulation.
>
> Once JPEG is adopted as the perturbation mechanism, it can no longer be treated as a fixed, standalone component. The perturbation depends directly on learnable JPEG parameters—most notably the quantization tables—which are optimized during training. Consequently, the perturbation process and JPEG configuration become inherently coupled: optimizing the perturbation necessarily entails updating the JPEG parameters.
>
> To further support the generality of our framework, Section A.7 shows that replacing JPEG with alternative differentiable perturbation modules—such as a **convolutional autoencoder**—while maintaining the same alternating optimization scheme yields comparable performance gains. This demonstrates that our framework and algorithm are not restricted to any specific codec but serve as a general optimization framework.
>
> >W3. Theoretical Limits. Convergence evidence (Fig. 5) is empirical; there is no formal rate or stationarity guarantee for the alternating updates. A brief analysis (e.g., monotonic ascent under bounded curvature, or conditions ensuring convergence to a critical point) would solidify the algorithmic contribution.
>
> **Reply**:  Thank you for raising this concern. As described in Section A.4 of the revised manuscript, if the impact of the random mini-batch sampling and SGD is ignored, the alternating algorithm is guaranteed to converge in theory, since for any given $w$, the inner minimization has an analytical solution. Although in practice the alternating algorithm may not converge to a global minimum, this is not the limitation of our algorithm, but the nature of all SGD-based deep learning algorithms.
>
> Beyond the theoretical analysis, we also provide empirical evidence of convergence on ImageNet by tracking the CMI value, the DJIP loss, and the total training objective. The CMI value increases steadily throughout training, and both the DJIP loss and the overall objective decrease smoothly. These observations indicate that the proposed alternating algorithm behaves stably in practice.
>
> >W4.  Data Efficiency. Experiments rely on high-quality data (e.g., PixMo-Cap); low-data regimes (few-shot/long-tail KD) remain untested. Evaluating DJIP under subsampling, noisy labels, or class imbalance—and reporting data-efficiency curves—would demonstrate robustness where KD is most needed.
>
> **Reply**:  Thank you for raising this important point. We would like to clarify that our experiments do **not** rely on high-quality datasets such as PixMo-Cap; instead, all evaluations are conducted on standard benchmarks—CIFAR-100 and ImageNet—which do not provide exceptionally curated annotations and are widely used for assessing KD methods. Therefore, DJIP does not depend on specialized high-quality data.
>
> We agree that few-shot and long-tail KD represent valuable and challenging settings where robustness and data efficiency are particularly important. While such experiments are beyond the current scope of this paper, we acknowledge that evaluating DJIP under low-data, noisy-label, and class-imbalanced scenarios constitutes a promising direction for future work. We appreciate your suggestions and plan to explore these regimes in the future.

---

> ### Author Response · Authors · 2025-11-25
> **Reply to Reviewer gUa4 (part 3)**
>
> ## Questions
> >Q1. Computational Overhead & Scaling. How does DJIP’s end-to-end cost scale with input resolution and batch size relative to vanilla KD? Please report latency (ms/img), throughput (img/s), and peak memory across GPU/CPU settings. Have you explored hardware-aware optimizations of the JPEG layer (e.g., fused CUDA kernels, SIMD/Neon intrinsics, TensorRT plugins), and what is the measured speedup?
>
> **Reply**: Thank you for the insightful questions. Regarding computational overhead, please refer to our reply to W1. In terms of **computational scaling**, the end-to-end cost of DJIP grows approximately **linearly with batch size** and **quadratically with input resolution** (i.e., proportional to the image area), which is consistent with the behavior of vanilla KD. This is because both the backbone forward pass and the differentiable JPEG layer operate on every spatial location, and thus their per-sample computational cost grows proportionally to the image area. Across the batch size, however, each sample is processed independently, so the total cost grows linearly with batch size. Consequently, DJIP preserves the same scaling behavior as vanilla KD in both axes.
>
> In practice, as shown in **Table A** and **Table B**, DJIP closely follows the scaling behavior of vanilla KD for both input resolution and batch size, with the only difference being a small constant overhead stemming from the JPEG module.
>
> **Table A**: Comparison of distillation complexity when scaling with batch size (BS) on VGG-13 $\Rightarrow$ VGG-8 pairs and CIFAR-100 dataset with vanilla KD method.
> | KD Method           | Throughput (img/ms) | Peak GPU Memory (MB) | Total Time (s) |
> |---------------------|---------------------|----------------------|----------------|
> | Vanilla KD (BS=32)  | 4.83    				    | 480                  | 2983.2         |
> | Vanilla KD (BS=64)  | 7.11    				    | 614                  | 2026.6         |
> | Vanilla KD (BS=128) | 8.85    				    | 840                  | 1627.2         |
> | DJIP (BS=32)        | 3.56   				 	  	| 482                  | 4048.8         |
> | DJIP (BS=64)        | 5.60   				 		  | 624                  | 2572.8         |
> | DJIP (BS=128)       | 7.19   				 		  | 850                  | 2001.6         |
>
> **Table B**: Comparison of distillation complexity when scaling with image resolution (IR) on VGG-13 $\Rightarrow$ VGG-8 pairs and CIFAR-100 dataset with vanilla KD method.
> | KD Method           | Throughput (img/ms) | Peak GPU Memory (MB) | Total Time (s) |
> |---------------------|---------------------|----------------------|----------------|
> | Vanilla KD (IR=32)  | 4.83    				    | 480                  | 2983.2         |
> | Vanilla KD (IR=64)  | 2.97    				    | 1292                 | 4840.8         |
> | Vanilla KD (IR=128) | 0.87    				    | 4434                 | 18410.4        |
> | DJIP (IR=32)        | 3.56   				 	  	| 482                  | 4048.8         |
> | DJIP (IR=64)        | 2.58   				 		  | 1498                 | 5575.2         |
> | DJIP (IR=128)       | 0.69   				 		  | 4800                 | 20841.6        |
>
> For the **second** question, the differentiable JPEG implementation inherits from `nn.Module` and therefore naturally supports CUDA GPU parallelism. Although we have not yet developed hardware-aware optimizations, these techniques are compatible with our design and represent promising avenues for further reducing latency. We therefore view such optimizations as future engineering extensions rather than limitations of DJIP’s current formulation.
>
>
> >Q2. Format Generalization. Beyond JPEG, have you evaluated differentiable alternatives (e.g., WebP/AVIF or learned compression) to test the mechanism’s generality? A head-to-head comparison controlling bitrate/PSNR would clarify whether gains stem from JPEG’s structure or from compression-induced perturbations more broadly.
>
> **Reply**:  Thank you for the insightful questions. Regarding format generalization, we would like to clarify that the proposed alternating optimization framework is fully agnostic to the choice of input perturbation mechanism. As stated in Theorem 1, the parameter $\omega$ represents any perturbation parameters and is not restricted to JPEG quantization tables. The differentiable JPEG module used in our experiments is merely one practical instantiation—selected for its simplicity, efficiency, and the availability of mature differentiable implementations. In principle, this perturbation layer can be replaced by any differentiable codec, including differentiable WebP/AVIF or learned compression models. Consistent with this interpretation, Section A.7 demonstrates that replacing JPEG with other differentiable perturbation mechanisms—such as a **convolutional autoencoder**—while keeping the alternating optimization unchanged, achieves similar performance improvements.

---

> > ### Author Response · Authors · 2025-11-25
> > **Reply to Reviewer gUa4 (part 4)**
> >
> > Regarding the second point, DJIP is **not** designed to optimize bitrate, PSNR, or any other compression-oriented metric. The perturbation is optimized exclusively to maximize the CMI value, with the goal of improving student performance, independent of perceptual quality or reconstruction fidelity. Therefore, head-to-head comparisons under matched bitrate or PSNR are not directly meaningful within our framework. Nonetheless, as shown in Figure 4, the CMI-driven optimization consistently learns a characteristic pattern: it applies a stronger perturbation to the luminance (Y) channel than to the chrominance (Cb/Cr) channels. This behavior differs markedly from traditional JPEG heuristics (which compress high frequencies more than low frequencies) and indicates that DJIP’s effectiveness arises from the CMI-driven perturbation mechanism itself, rather than from JPEG-specific structural properties.
> >
> >
> > >Q3. Algorithmic Stability. Does the alternating scheme exhibit instability when class centroids shift quickly (e.g., early training, long-tail classes)? Would momentum/EMA, trust-region or proximal updates, or line-search on the CMI objective improve stability and convergence consistency? Any failure cases or sensitivity analyses?
> >
> > **Reply**: Thank you for the thoughtful question regarding the stability of the alternating optimization scheme. For CIFAR-100 and ImageNet, both of which are balanced datasets, class centroids do not exhibit long-tail instability issues. Moreover, as shown in Figure 6, because DJIP trains the JPEG layer based on a _well-trained_ teacher model, the class-conditional feature distributions remain stable throughout training, so the centroids do not shift rapidly in practice.
> >
> > For ImageNet, we additionally use a mini-dataset to estimate the centroids. In this setting, sampling noise may cause the centroids to fluctuate more noticeably. To mitigate this, we adopt an EMA update strategy for centroid estimation, which significantly stabilizes the alternating optimization. The primary failure case occurs when centroid estimation becomes inaccurate due to insufficient update frequency, as analyzed in **Table 10** of the revised manuscript; this can temporarily weaken the CMI guidance and cause performance degradation.
> >
> > **Table 10**: Effect of the centroid update interval in DJIP on CIFAR-100 dataset. Interval denotes the number of steps between two centroid updates.
> > | Teacher   | Student | Method      | CMI   | KD    | DKD   | DIST  | CC    | RKD   |
> > |-----------|---------|-------------|-------|-------|-------|-------|-------|-------|
> > | ResNet-50 | VGG-8   | CE          | 0.009 | 73.81 | 73.94 | 74.11 | 70.25 | 71.50 |
> > | ResNet-50 | VGG-8   | Interval=3  | 0.341 | 74.49 | 75.87 | 74.58 | 70.92 | 71.87 |
> > | ResNet-50 | VGG-8   | Interval=5  | 0.341 | 74.48 | 75.87 | 74.80 | 70.92 | 71.93 |
> > | ResNet-50 | VGG-8   | Interval=50 | 0.341 | 74.33 | 75.58 | 74.63 | 70.78 | 71.81 |
> > | ResNet-50 | VGG-8   | Interval=500| 0.342 | 74.32 | 75.49 | 74.52 | 70.68 | 71.73 |
> >
> >
> > >Q4. Black-Box Teachers. How does DJIP perform when the teacher is accessible only via queries (API/closed-source), precluding gradient flow? Can zeroth-order/finite-difference estimates, score-matching proxies, or pseudo-label distillation approximate the CMI objective, and with what accuracy–cost trade-offs?
> >
> > **Reply**: We thank you for the insightful comment on extending DJIP to the black-box teacher setting. DJIP is primarily designed as a **white-box** method. However, if the output probability distribution of the teacher model is accessible (e.g., through an API), our framework can still be applied. As discussed in Section 3.2, we treat the classification DNN as a fixed mapping function, which effectively operates like a **black box**. In this case, our proposed alternating algorithm for maximizing CMI remains valid. For scenarios where gradient access is unavailable, the zeroth-order or finite-difference estimation methods suggested by you can indeed be adopted to update the trainable input perturbation parameters (e.g., the JPEG quantization parameters in DJIP). We appreciate this valuable suggestion and will further explore this potential extension in those directions.
> >
> > **We thank you once again for the time and effort you have dedicated to reviewing our work. If our responses satisfactorily address all the concerns you raised, we would be grateful if you could consider raising the score to a higher level.**

---

> ### Author Response · Authors · 2025-11-29
> **Reply to Reviewer gUa4 (part 5)**
>
> >3）Although the convergence analysis in Section A.4 is based on simplifying assumptions (e.g., ignoring SGD stochasticity), and the empirical results on ImageNet do show stable CMI growth and decreasing loss, I remain somewhat concerned about the theoretical guarantees. The clarification that EMA is used for centroid updates on ImageNet helps, but it does not fully eliminate my worries about stability at scale.
>
> **Reply**: Thank you for the follow-up question to the previous Q3. In Section 4.2, our objective function is defined as $L_{CE} - \lambda L_{DJIP}$. Here, $L_{CE}$ depends only on the JPEG parameters $w$, while $L_{DJIP}$ depends on both $w$ and the proposed "backward channel" $Q(\cdot \mid i, y)$. As discussed in Section A.4, after substituting the analytic solution of the inner minimization for $Q(\cdot \mid i, y)$, the objective function depends only on $w$, which can be optimized directly using **standard SGD**.
>
> At each update step, the minibatch loss serves as an unbiased estimator of the full-dataset loss. By the **law of large numbers**, the minibatch estimate converges to the population loss as the batch size increases. In our ImageNet experiments, we use a batch size of 256, which is considered sufficiently large in practice to ensure a low-variance gradient estimate. As a result, the optimization trajectory closely tracks that of full-batch gradient descent, and the training dynamics remain **stable**.
>
> In the DJIP setting, **several additional properties** contribute to the stability of the training dynamics: (1) the teacher model is pretrained and kept fixed, thus the class centroids do not shift rapidly throughout training; (2) the mini-dataset used for centroid computation is sampled once and remains fixed throughout training; (3) this mini-dataset is class-balanced; (4) mini-dataset is sufficiently large (16,000 images); and (5) the EMA updates applied provide additional smoothing, reducing short-term fluctuations in the centroid estimates. Together, these factors yield a stable loss decrease and monotonic CMI growth.
>
> Moreover, the convergence behavior of SGD under mild minibatch noise is **well-established** in the optimization literature [1, 2], forming a core theoretical basis for modern deep learning. Since our method reduces exactly to this standard setting after the inner step is analytically solved, it does **not introduce any additional instability** beyond that of ordinary SGD. The empirical results on both ResNet-34 and ResNet-152 in Section A.4 further confirm that DJIP operates in a stable and predictable regime **at scale**.
>
> **Reference**:
>
> [1] Hardt, Moritz, Benjamin Recht, and Yoram Singer. *Train faster, generalize better: Stability of stochastic gradient descent.* arXiv preprint arXiv:1509.01240, 2015.
>
> [2] Ghadimi, Saeed, and Guanghui Lan. *Stochastic First- and Zeroth-order Methods for Nonconvex Stochastic Programming.* arXiv preprint arXiv:1309.5549, 2013.

---

> ### Author Response · Authors · 2025-11-29
> **Reply to Reviewer gUa4 (part 6)**
>
> >4）I understand the authors point that standard KD benchmarks such as CIFAR and ImageNet are the main focus here but I still have concerns about the lack of evaluation in low data settings although this may be left for future work it remains an important stress test that would strengthen the overall validation of the method.
>
> **Reply**: Thank you for raising this important point regarding evaluation under limited-data conditions. We agree that **low-data** scenarios constitute a meaningful stress test for knowledge distillation methods. To address this concern, we provide an additional experiment inspired by the standard **few-shot classification** protocol.
>
> Following the few-shot setting [1], only a $\beta$ % subset of samples per class is used for training the student, while the JPEG layer and the DJIP teacher are trained on the full dataset. We conduct experiments on CIFAR-100 using the WRN-40-2 → WRN-16-2 teacher–student pair and evaluate $\beta \in $ {5,10,20,50,75} under vanilla KD. The results are summarized in **Table 13**.
>
> As shown, the student consistently benefits from distillation with the DJIP teacher across all levels of data scarcity. The gains are particularly pronounced in the most challenging regimes (e.g., +5.81% at $\beta = 5$), demonstrating that DJIP provides more informative supervision than standard KD when the student receives substantially fewer samples. These findings further confirm the **robustness and generalizability** of our method under constrained-data settings.
>
> **Table 13**: Comparison of student accuracy (%) under few-shot vanilla KD distillation settings on CIFAR-100, while only a $\beta$ percent subset of samples per class is used to train the student.
> | Teacher  | Student  | Method   |$\beta$=5 |$\beta$=10|$\beta$=20|$\beta$=50|$\beta$=75|
> |----------|----------|----------|-------|-------|-------|-------|-------|
> | WRN-40-2 | WRN-16-2 | CE       | 34.75 | 50.73 | 60.93 | 69.60 | 72.17 |
> | WRN-40-2 | WRN-16-2 | DJIP     | 40.56 | 53.95 | 62.98 | 70.87 | 73.15 |
> | WRN-40-2 | WRN-16-2 | $\Delta$ | +5.81 | +3.22 | +2.06 | +1.27 | +0.98 |
>
> **Reference**:
> [1] Luo, Xu, et al. *A closer look at few-shot classification again.* International Conference on Machine Learning. PMLR, 2023.

---

### Official Review · Reviewer_RUYs · 2025-10-31

**Soundness:** 3
**Presentation:** 3
**Contribution:** 3
**Rating:** 6
**Confidence:** 4

**Summary:**

The paper introduces Differentiable JPEG-based Input Perturbation (DJIP), a novel framework designed to enhance teacher-student knowledge transfer in knowledge distillation (KD) without modifying the teacher model.
DJIP uses a differentiable JPEG layer that perturbs the teacher’s inputs to directly increase conditional mutual information (CMI), which improves distillation effectiveness.
The paper also proposes a new alternating optimization algorithm to efficiently learn the JPEG layer's coding parameters to maximize the perturbed CMI.
Extensive experiments on CIFAR-100 and ImageNet demonstrate that DJIP consistently boosts student accuracy by up to 4.11% while being computationally efficient and compatible with standard KD pipelines.

**Strengths:**

1. The paper proposes a novel technique, Differentiable JPEG-based Input Perturbation (DJIP), which is a plug-and-play framework for improving knowledge distillation (KD) without requiring modifications to the teacher model.

2. The method is extensively evaluated on CIFAR-100 and ImageNet datasets, demonstrating consistent improvements in student accuracy, with gains up to 4.11%.

3. DJIP is computationally lightweight and integrates seamlessly with standard KD pipelines, making it an efficient solution for enhancing knowledge transfer. The method optimizes just the JPEG layer without modifying the teacher model, ensuring low overhead.

4. The proposed method works well across both same-architecture and cross-architecture (CNN-to-ViT) settings, which shows that DJIP can be applied broadly in knowledge distillation tasks, including distilling between heterogeneous model types.

5. DJIP is orthogonal to other state-of-the-art methods like MCMI, as demonstrated by the paper's results. This suggests that DJIP can be integrated with existing KD techniques to further enhance performance, providing flexibility in distillation pipelines.

**Weaknesses:**

1. While the method is effective in practice, the paper provides limited theoretical analysis of why perturbing the input via the JPEG layer improves distillation beyond just the CMI maximization objective.

2. The method heavily relies on a differentiable JPEG layer, which could limit its applicability to certain use cases or architectures where JPEG compression may not be optimal or desirable.

3. The alternating optimization algorithm introduced for learning the JPEG coding parameters adds a layer of complexity. While efficient, the algorithm may require fine-tuning, and its performance could vary depending on the choice of hyperparameters, such as lambda.

4. The usage of JPEG limits the method within the image domain. The method’s effectiveness might degrade on different types of data, especially those that are not image-based or have different structures.

**Questions:**

1. How does DJIP compare to other input perturbation methods in terms of generalization to different types of data, such as non-image datasets or tasks beyond image classification? Are there any potential limitations or challenges when applying DJIP to these domains?

2. The proposed alternating optimization algorithm is central to maximizing the perturbed CMI. Could you provide more details on how this algorithm scales with larger models or more complex datasets, and whether there are any concerns regarding its stability or convergence in these cases?

---

> ### Author Response · Authors · 2025-11-25
> **Reply to Reviewer RUYs (part 1)**
>
> Dear Reviewer **RUYs**:
>
> We thank you for taking the time to review our paper and for providing valuable feedback. Your summary is accurate and well-captured. Below, please find our responses to your insightful comments.
>
> ## Weaknesses
> > W1. While the method is effective in practice, the paper provides limited theoretical analysis of why perturbing the input via the JPEG layer improves distillation beyond just the CMI maximization objective.
>
> **Reply**: Thank you for raising this important question. Our design is grounded in the theoretical foundation established by MCMI (Ye *et al.*, 2024), which shows that maximizing the CMI $I(X; \hat{Y} \mid Y)$ leads to a more accurate estimation of the unknown Bayes conditional probability distribution (BCPD). Prior analyses [1, 2, 3] collectively demonstrate that the closer the teacher’s prediction is to the BCPD, the lower the variance of the student’s objective and thereby the better the student’s performance would be. The Eigen-CAM figures in Section A.10 of MCMI also provide a visualization of the influence on feature space when CMI is maximized. The CMI maximized output supervision signals are able to offer more "dark knowledge" from the background to students. Similar perspectives have also been explored in recent KD studies [4, 5].
>
> More formally, Proposition 1 in MCMI proves that  $$ I(X;\hat{Y} \mid Y) \le I(X; \mathcal{f}_X \mid Y), $$  where $\mathcal{f}_X$ denotes the teacher’s feature maps. Thus, increasing the perturbed CMI implicitly increases the upper bound of information transmitted through the teacher’s feature hierarchy. Since DJIP operates solely in the input space, its influence propagates throughout the network, enabling the student to receive richer and more informative signals.
>
> **References:**
> [1] Menon, A. K., Rawat, A. S., Reddi, S., Kim, S., & Kumar, S. (2021). *A statistical perspective on distillation*. In *Proceedings of the 38th International Conference on Machine Learning (ICML 2021)*, pp. 7632–7642. PMLR.
>
> [2] Tri Dao, Govinda M Kamath, Vasilis Syrgkanis, and Lester Mackey. Knowledge distillation as semiparametric inference. In International Conference on Learning Representations, 2020.
>
> [3] En-Hui Yang, Shayan Mohajer Hamidi, Linfeng Ye, Renhao Tan, and Beverly Yang. Conditional mutual information constrained deep learning for classification. arXiv preprint arXiv:2309.09123, 2023.
>
> [4] Fang, L., Chen, Y., Zhong, W., & Ma, P. (2024). *Bayesian Knowledge Distillation: A Bayesian Perspective of Distillation with Uncertainty Quantification*. In ICML, 12935–12956.
>
> [5] Li, W., Li, L., Lee, M. G., Sun, S., Zhang, L., Xue, W., & Guo, Y. (2025). *BayesKD: Bayesian Knowledge Distillation for Compact LLMs in Constrained Fine-tuning Scenarios*. In _Findings of ACL 2025_, 138–152.
>
> >W2. The method heavily relies on a differentiable JPEG layer, which could limit its applicability to certain use cases or architectures where JPEG compression may not be optimal or desirable.
>
> **Reply**: We appreciate your concern. We would like to clarify that while this paper presents DJIP using JPEG as the perturbation mechanism, the underlying framework is **generic** and extends to any form of teacher input perturbation. The theoretical results—Theorem 1 and the alternating optimization algorithm—remain valid regardless of the specific perturbation method. Our method **does not** fundamentally rely on JPEG itself. As stated in Theorem 1, the parameter $\omega$ represents **any** perturbation parameter and is not restricted to JPEG quantization tables. The differentiable JPEG module used in our experiments is merely one practical instantiation—selected for its simplicity, efficiency, and the availability of mature differentiable implementations. In principle, this perturbation layer can be replaced by any differentiable codec.
>
> Consistent with this interpretation, Section A.7 of the revised manuscript demonstrates that replacing JPEG with other differentiable perturbation mechanisms—such as a **convolutional autoencoder**—while keeping the alternating optimization unchanged, achieves similar performance improvements.

---

> ### Author Response · Authors · 2025-11-25
> **Reply to Reviewer RUYs (part 2)**
>
> >W3. The alternating optimization algorithm introduced for learning the JPEG coding parameters adds a layer of complexity. While efficient, the algorithm may require fine-tuning, and its performance could vary depending on the choice of hyperparameters, such as lambda.
>
> **Reply**: Thank you for your insightful comment. We would like to offer two clarifications.
>
> First, regarding the hyperparameter such as $\lambda$, we conducted an ablation study, detailed in **Table 7** of the revised manuscript. The results show that the student’s Top-1 accuracy follows a quasiconcave trend as $\lambda$ varies, with peak performance at $\lambda = 1.4$. More importantly, across a wide range of $\lambda$ values, the student consistently outperforms the baseline. This demonstrates the robustness of our method to the choice of $\lambda$, and our default setting is already close to optimal.
>
> **Table 7**: Effect of $\lambda$ in DJIP on ImageNet dataset and ResNet-34 $\rightarrow$ ResNet-18 pairs.
> | KD              | CE     |λ = 0.4 |λ = 1.0 | λ = 1.4    |λ = 2.0 |λ = 2.4 | λ = 3.0 | λ = 4.0 |
> |-----------------|--------|--------|--------|------------|--------|--------|---------|---------|
> | Teacher CELoss  | 0.560  | 0.568  | 0.568  | 0.582      | 0.630  | 0.595  | 0.627   | 0.871   |
> | Teacher CMI     | 0.7180 | 0.7257 | 0.7268 | 0.7382     | 0.7869 | 0.7523 | 0.7825  | 0.9866  |
> | Student Acc (%) | 70.660 | 71.362 | 71.452 | **71.654** | 71.274 | 71.370 | 71.290  | 70.694  |
>
> For other ablation studies on hyperparameters involved in training the JPEG layer, including learning rate, we kindly refer you to Section A.6 of the revised manuscript.
>
> Second, we did not perform any additional hyperparameter tuning for the KD methods used during the online distillation stage. For fairness and reproducibility, we strictly followed the official training recipes reported in the corresponding papers.
>
>
> >W4. The usage of JPEG limits the method within the image domain. The method’s effectiveness might degrade on different types of data, especially those that are not image-based or have different structures.
>
> **Reply**: Thank you for raising this important point. Both JPEG and our DJIP framework are primarily designed for vision-based tasks, and we understand your interest in potential extensions to broader domains such as LLMs (e.g., LLaMA) or generative models (e.g., Stable Diffusion). Theoretically, DJIP has the potential to generalize beyond classification, since the core alternating optimization algorithm does not rely on image-specific assumptions. However, several non-trivial challenges remain unsolved for these domains.
>
> **First**, it is unclear how to define or construct _differentiable input perturbation mechanisms_ suitable for text or token sequences. Potential ways could be explored, but each presents non-trivial limitations. For example, one might consider (i) **perturbing continuous token embeddings** through adversarial or learnable noise (e.g., FGSM/PGD-style updates)[1, 2], (ii) **optimizing soft prompts or continuous prefix vectors** as surrogate perturbations [3, 4], or (iii) **using a learnable paraphrasing module** (e.g., a small seq2seq network [5]) to generate parametric, differentiable approximations of discrete token edits. While these techniques enable gradient-based updates, they do not translate cleanly to discrete token spaces and often break alignment between perturbed embeddings and actual text realizations. Moreover, unlike images, where perturbations remain within the same perceptual manifold, text perturbations frequently introduce semantic drift, length or grammatical inconsistencies, making reliable and stable optimization substantially more difficult. As a result, extending DJIP to the textual domain would require additional theoretical and algorithmic innovations to preserve semantic fidelity while supporting end-to-end differentiability.
>
>
> **Second**, the definition of _conditional mutual information (CMI)_ and the construction of meaningful _cluster centroids_ for LLMs is substantially more complex than in vision tasks. In classification-based vision models, class labels naturally induce a well-defined partition of the representation space, enabling centroid estimation through supervised feature aggregation. This characteristic still exists in vision segmentation tasks. In contrast, LLMs operate on high-dimensional token trajectories and support a wide range of open-ended generation behaviors, making it unclear how to define discrete “clusters’’ that correspond to meaningful conditional variables. Possible proxies—such as grouping by **next-token distributions**[6], **semantic intent**[7], or **latent discourse states**[8]—remain unstable and highly sensitive to prompt structure. Furthermore, CMI in generative models depends on the joint distribution over entire output sequences rather than a fixed label, which complicates both its estimation and its maximization.

---

> ### Author Response · Authors · 2025-11-25
> **Reply to Reviewer RUYs (part 3)**
>
> These challenges imply that adapting DJIP to LLMs would require more insights on cluster structure and input perturbation methods. We regard above directions as highly promising but beyond the scope of the current paper.
>
> **References:**
>
> [1] Sato, M., Suzuki, J., Shindo, H., & Matsumoto, Y. (2018). *Interpretable Adversarial Perturbation in Input Embedding Space for Text*. In _IJCAI-ECAI 2018_. arXiv:1805.02917.
>
> [2] Yuan, L., Zhang, Y., Chen, Y., & Wei, W. (2023). *Bridge the Gap Between CV and NLP! A Gradient-based Textual Adversarial Attack Framework*. In A. Rogers, J. Boyd-Graber, & N. Okazaki (Eds.), _Findings of the Association for Computational Linguistics: ACL 2023_, 7132–7146. Toronto, Canada: Association for Computational Linguistics.
>
> [3] Petrov, A., Torr, P. H. S., & Bibi, A. (2023). *When Do Prompting and Prefix-Tuning Work? A Theory of Capabilities and Limitations*. arXiv:2310.19698.
>
> [4] Li, Z., Liu, Y., Su, Y., & Collier, N. (2025). "Prompt Compression for Large Language Models: A Survey". In L. Chiruzzo, A. Ritter, & L. Wang (Eds.), _Proceedings of the 2025 Conference of the Nations of the Americas Chapter of the Association for Computational Linguistics: Human Language Technologies (NAACL 2025, Volume 1: Long Papers)_, 7182–7195. Albuquerque, New Mexico: Association for Computational Linguistics.
>
> [5] Chen, W., Tian, J., Xiao, L., He, H., & Jin, Y. (2020). *A Semantically Consistent and Syntactically Variational Encoder-Decoder Framework for Paraphrase Generation*. In D. Scott, N. Bel, & C. Zong (Eds.), _Proceedings of the 28th International Conference on Computational Linguistics (COLING 2020)_, 1186–1198. Barcelona, Spain (Online): International Committee on Computational Linguistics.
>
> [6] Zhao, Y., Behnia, T., Vakilian, V., & Thrampoulidis, C. (2024). "Implicit Geometry of Next-token Prediction: From Language Sparsity Patterns to Model Representations". arXiv:2408.15417.
>
> [7] Liu, J., Shang, Z., Ke, W., Wang, P., Luo, Z., Liu, J., Li, G., & Li, Y. (2025). "LLM-Guided Semantic-Aware Clustering for Topic Modeling". In W. Che, J. Nabende, E. Shutova, & M. T. Pilehvar (Eds.), _Proceedings of the 63rd Annual Meeting of the Association for Computational Linguistics (ACL 2025, Volume 1: Long Papers)_, 18420–18435. Vienna, Austria: Association for Computational Linguistics.
>
> [8] Saglam, B., Kassianik, P., Nelson, B., Weerawardhena, S., Singer, Y., & Karbasi, A. (2025). "Large Language Models Encode Semantics in Low-Dimensional Linear Subspaces". arXiv:2507.09709.
>
> ## Questions
> >Q1. How does DJIP compare to other input perturbation methods in terms of generalization to different types of data, such as non-image datasets or tasks beyond image classification? Are there any potential limitations or challenges when applying DJIP to these domains?
>
> **Reply**: Thank you for bringing up this point. Regarding the first question, the generalization of our method can be understood at two different levels.
>
> **First**, when DJIP is interpreted strictly as a _JPEG-based perturbation technique_, it is indeed similar to CKD, TALD, and other image-specific perturbation approaches. These methods rely on vision-domain priors—such as frequency decomposition or spatial augmentations—and therefore do not directly transfer to non-image modalities. Under this view, DJIP remains broadly applicable within the image domain, supporting tasks beyond classification (e.g., segmentation).
>
> **Second**, if we step back and consider the _alternating optimization framework_ that forms the foundation of DJIP, the situation is different. The core idea—embodied in Theorem 1 and the alternating CMI-maximization algorithm—is inherently modality-agnostic. This framework does not depend on JPEG or even visual data. In principle, it can be instantiated with suitable perturbation operators for other modalities, including text, as we discussed earlier in our response to Comment W4.
>
> Regarding the second question about potential limitations or challenges, we refer you to our response to Comment W4 above.
>
> >Q2. The proposed alternating optimization algorithm is central to maximizing the perturbed CMI. Could you provide more details on how this algorithm scales with larger models or more complex datasets, and whether there are any concerns regarding its stability or convergence in these cases?
>
> **Reply**: For the **first** question, CIFAR-100 is a widely used dataset comprising 60,000 color images of resolution 32 × 32 pixels, categorized into 100 classes, while ImageNet is a large-scale dataset containing approximately 1.2 million training images of size 224 × 224. When scaling our method from CIFAR-100 to ImageNet, we made several modifications to ensure computational feasibility and stable centroid estimation, as detailed in Section A.5 of the revised manuscript.

---

> ### Author Response · Authors · 2025-11-25
> **Reply to Reviewer RUYs (part 4)**
>
> For **CIFAR-100**, the class centroid is computed over the _entire training set_. Updating this centroid after every iteration is unnecessary and computationally wasteful. As analyzed in Section A.6.4, increasing the update frequency does not further improve student performance. Therefore, we update the centroid **once every five outer-minimization steps** (i.e., every 5 iterations), which provides an effective trade-off between stability and efficiency.
>
> For **ImageNet**, directly computing the centroid over the full training set is impractical due to its scale. To address this, before training begins, we construct a **fixed and balanced sub-dataset** of 16 randomly selected images per class (16,000 images in total). All centroid updates are then performed exclusively on this sub-dataset with an update frequency of **one** (i.e., a centroid update at every outer iteration). To ensure additional robustness, we further apply an EMA with momentum 0.9 to smooth the centroid evolution.
>
> Across both datasets, the **teacher model is pretrained and fixed**, ensuring that the underlying feature distribution remains stable. Consequently, the class centroids do not shift rapidly during training, which helps maintain the stability of the alternating optimization.
>
> For the **second** question regarding **convergence**, as shown in Section A.4 of the revised manuscript, if the impact of the random mini-batch sampling and SGD is ignored, the alternating algorithm is guaranteed to converge in theory, since for any given $w$, the inner minimization has an analytical solution. Although in practice, the alternating algorithm may not converge to a global minimum. This is not the limitation of our algorithm, but the nature of all SGD-based deep learning algorithms.
>
> Beyond this theoretical analysis, we also provide empirical evidence of convergence on ImageNet by tracking the CMI value, the DJIP loss, and the total training objective. The CMI value increases steadily throughout training, and both the DJIP loss and the overall objective decrease smoothly. These observations indicate that the proposed alternating algorithm behaves stably in practice. Consequently, we do not observe any stability or convergence issues when scaling from CIFAR-100 to ImageNet.
>
> **We thank you once again for the time and effort you have dedicated to reviewing our work. If our responses satisfactorily address all the concerns you raised, we would be grateful if you could consider raising the score to a higher level.**

---

### Official Review · Reviewer_tbBh · 2025-11-01

**Soundness:** 3
**Presentation:** 3
**Contribution:** 2
**Rating:** 4
**Confidence:** 4

**Summary:**

This paper proposes Differentiable JPEG-based Input Perturbation (DJIP), a plug-and-play framework for improving knowledge distillation (KD) without modifying the teacher model.
DJIP introduces a differentiable JPEG layer before the teacher network to perturb inputs in a way that maximizes conditional mutual information (CMI) between inputs and outputs.
The authors also design an alternating optimization algorithm that jointly optimizes cross-entropy and CMI losses, enabling efficient learning of JPEG quantization parameters.
Extensive experiments on CIFAR-100 and ImageNet show consistent student accuracy improvements (up to 4.11%) across CNNs and ViTs, and compatibility with many KD baselines.

**Strengths:**

1.	Solid theoretical foundation: The CMI-based alternating optimization is mathematically consistent and improves upon fixed-centroid MCMI.
2.	Comprehensive empirical validation: Tested on multiple datasets and KD frameworks, with consistent improvements.
3.	Orthogonality: Demonstrated compatibility with both MCMI and CKD, suggesting general utility according to the paper.
4.	Reproducibility: Implementation details and appendices are complete and transparent.

**Weaknesses:**

1.	Limited theoretical novelty: The alternating CMI formulation is an incremental improvement over prior MCMI, and the key novelty lies in engineering implementation (JPEG layer).
2.	Insufficient qualitative analysis: The paper lacks visualization or interpretation of how and why JPEG perturbations affect teacher responses.
3.	Limited performance improvements: As an additional, plug-and-play module, DJIP offers limited performance enhancements to networks, which focuses more on engineering optimizations than theoretical innovations.

**Questions:**

1.	Could you provide a runtime and GPU memory usage of training DJIP layer?
2.	How sensitive is the method to the hyperparameter λ? Are there any cases where maximizing CMI harms student performance?
3.	Did the authors observe any instability during training due to learning rate choice? How robust is DJIP to different optimizer configurations (e.g., step size, momentum, or learning)
4.	Have you considered using other differentiable codecs (e.g., differentiable WebP or learned compression networks) as a generalization of DJIP?

---

> ### Author Response · Authors · 2025-11-25
> **Reply to Reviewer tbBh (part 1)**
>
> Dear Reviewer **tbBh**:
>
> We thank you for taking the time to review our paper and for providing valuable feedback. Below we present our responses to your comments.
>
>
> ## Weaknesses
> > W1.  Limited theoretical novelty: The alternating CMI formulation is an incremental improvement over prior MCMI, and the key novelty lies in engineering implementation (JPEG layer).
>
> **Reply**: We appreciate your comment and would like to clarify a potential misunderstanding. While our work is inspired by MCMI (Ye _et al._, 2024), the proposed DJIP framework is built upon a fundamentally different optimization problem and objective design. Specifically, MCMI maximizes CMI by updating the **model parameters** and relies on a **fixed-centroid approximation** when computing the CMI term in its objective $\mathcal{L}_{\text{MCMI}} = CE - \lambda \times CMI$. As discussed in Section A.1, this approximation does not fully resolve the underlying CMI maximization problem.
>
> In contrast, DJIP keeps the teacher networks **entirely fixed** and performs CMI optimization purely through perturbation in the **input-space**. By reformulating the CMI objective as a double minimization problem and optimizing it via alternating updates on the learnable perturbation parameters and the introduced **"backward channel"**, our loss becomes
> $\mathcal{L} = CE - \lambda \times \text{DJIP Loss}$.
> This design directly overcomes the limitations of MCMI’s fixed-centroid formulation. The introduction of the dummy "backward channel"—together with the alternating optimization procedure—is essential, as it allows DJIP to dynamically align the input perturbations with the evolving representation geometry of the teacher, while keeping the model itself unchanged.
>
> Importantly, the proposed alternating optimization scheme is **not tied to JPEG**. As stated in Theorem 1, the perturbation parameter $\omega$ is general and is _not_ restricted to JPEG quantization tables. The differentiable JPEG module used in our experiments serves only as a convenient instantiation due to its simplicity and widespread differentiable implementations. In principle, the perturbation layer can be replaced with any differentiable codec. Consistently, Section A.7 shows that substituting JPEG with other differentiable perturbation mechanisms—such as a **convolutional autoencoder**—while keeping the alternating optimization unchanged, yields similar performance gains.
>
> Therefore, DJIP extends beyond engineering implementation. It provides a new theoretical perspective by demonstrating that CMI can be effectively maximized in the input space through an alternating minimization framework that applies broadly across diverse differentiable perturbation mechanisms.
>
>
>
> >W2. Insufficient qualitative analysis: The paper lacks visualization or interpretation of how and why JPEG perturbations affect teacher responses.
>
> **Reply**: For **_how_** JPEG perturbations affect teacher responses, we refer you to Figures 6 and 7 in Section A.10 of the revised manuscript, where detailed visualizations illustrate how perturbations modify the teacher’s output probability distributions across different epochs.
>
> Regarding **_why_** input perturbations influence teacher responses, our design is grounded in the theoretical foundation established by MCMI, which shows that maximizing the CMI $I(X;\hat{Y}\mid Y)$ yields a more accurate approximation of the unknown Bayes conditional probability distribution (BCPD). Prior analyses [1–3] demonstrate that the closer the teacher’s prediction is to the BCPD, the lower the variance of the student’s learning objective becomes, thereby facilitating more stable and effective distillation. The Eigen-CAM figures in Section A.10 of MCMI also provide visualization of the influence on feature space when CMI is maximized. The CMI maximized output supervision signals are able to provide more "dark knowledge" from the background to students. Similar perspectives are further supported by recent KD studies [4, 5].
>
> More formally, _Proposition 1_ in MCMI establishes that $$ I(X;\hat{Y} \mid Y) \le I(X; \mathcal{f}_X \mid Y), $$ where $\mathcal{f}_X$ denotes the teacher’s feature maps. This implies that increasing the perturbed CMI implicitly increases the upper bound of information propagated through the teacher’s feature hierarchy. Although DJIP perturbs only the input, its effect propagates through all subsequent layers, enabling the student to receive richer and more informative supervisory signals. This provides a theoretical explanation for why optimized perturbations enhance the distillation process.
>
>
> **References:**
> [1] Menon, A. K., Rawat, A. S., Reddi, S., Kim, S., & Kumar, S. (2021). *A statistical perspective on distillation*. In *Proceedings of the 38th International Conference on Machine Learning (ICML 2021)*, pp. 7632–7642. PMLR.

---

> ### Author Response · Authors · 2025-11-25
> **Reply to Reviewer tbBh (part 2)**
>
> [2] Tri Dao, Govinda M Kamath, Vasilis Syrgkanis, and Lester Mackey. *Knowledge distillation as semiparametric inference*. In International Conference on Learning Representations, 2020.
>
> [3] En-Hui Yang, Shayan Mohajer Hamidi, Linfeng Ye, Renhao Tan, and Beverly Yang. *Conditional mutual information constrained deep learning for classification*. arXiv preprint arXiv:2309.09123, 2023.
>
> [4] Fang, L., Chen, Y., Zhong, W., & Ma, P. (2024). *Bayesian Knowledge Distillation: A Bayesian Perspective of Distillation with Uncertainty Quantification*. In ICML, 12935–12956.
>
> [5] Li, W., Li, L., Lee, M. G., Sun, S., Zhang, L., Xue, W., & Guo, Y. (2025). *BayesKD: Bayesian Knowledge Distillation for Compact LLMs in Constrained Fine-tuning Scenarios*. In _Findings of ACL 2025_, 138–152.
>
>
> >W3. Limited performance improvements: As an additional, plug-and-play module, DJIP offers limited performance enhancements to networks, which focuses more on engineering optimizations than theoretical innovations.
>
> **Reply**: Thank you for raising this point. We would like to clarify several important aspects:
>
> **First**, as noted in our response to Comment W1, DJIP goes well beyond engineering optimization. The framework includes substantial theoretical components—most notably Theorem 1 and the alternating optimization algorithm—which form the foundation of the entire method. Without these theoretical developments, the engineering components and subsequent empirical evaluations would not be possible.
>
> **Second**, the magnitude of performance enhancements should be judged within the context. DJIP is designed as an **orthogonal** framework, not as a replacement for existing KD strategies. Across all major KD categories—logit-based, feature-based, relation-based, and student-oriented—DJIP consistently improves the performance of strong baselines. Several of the baselines we enhance are already state-of-the-art under the same model–dataset configuration, yet DJIP still yields measurable and reproducible gains. Notably, **Table 4** of the original manuscript reports a 4.11% absolute improvement, which is widely regarded as significant in the KD literature.
>
> **Third**, while this paper presents DJIP using JPEG as the perturbation mechanism, the underlying framework is **generic** and extends to any form of teacher input perturbation (e.g., autoencoder-based transformations). The theoretical results—Theorem 1 and the alternating optimization algorithm—remain valid regardless of the specific perturbation method. JPEG is chosen for its simplicity and efficiency, but stronger perturbation strategies may yield even larger improvements.
>
> Overall, DJIP provides stable and non-trivial performance improvements across diverse KD categories and strong baselines, supported by a clear theoretical foundation and a lightweight optimization structure that distinguishes it from prior orthogonal approaches.
>
>
> ## Questions
> >Q1. Could you provide a runtime and GPU memory usage of training DJIP layer?
>
> **Reply**: Thank you for asking this question. As shown in the **Table C**, we have included the total runtime and GPU memory usage of the offline JPEG training stage. The experiment is conducted on 4 NVIDIA RTX A5000 GPUs on ImageNet.
>
> **Table C**: Training complexity of the JPEG layer measured on ResNet-34.
> | Teacher   | Runtime (s) | Peak GPU Memory (MB/GPU) |
> |-----------|-------------|--------------------------|
> | ResNet-34 | 43436.4     | 7750                     |
>
>
>
> >Q2. How sensitive is the method to the hyperparameter λ? Are there any cases where maximizing CMI harms student performance?
>
> **Reply**: Thank you for the thoughtful question. As shown in the ablation study in **Table 7** of the revised manuscript, the student’s Top-1 accuracy exhibits a quasiconcave trend as the hyperparameter $\lambda$ varies, with the best performance achieved at $\lambda = 1.4$. Across a broad range of $\lambda$ values, DJIP consistently outperforms the baseline, indicating that the method is reasonably robust and that our default setting is near-optimal.
>
> That said, we do observe failure cases when $\lambda$ becomes excessively large, such as $\lambda =4.0$ in **Table 7**. In such regimes, the optimization places disproportionate emphasis on maximizing CMI, leading to overly aggressive perturbations and degradation of the student’s accuracy. This confirms that while increasing CMI is generally beneficial, it must be balanced with maintaining meaningful input information for effective knowledge transfer.

---

> > ### Author Response · Authors · 2025-11-25
> > **Reply to Reviewer tbBh (part 3)**
> >
> > **Table 7**: Effect of $\lambda$ in DJIP on ImageNet dataset and ResNet-34 $\rightarrow$ ResNet-18 pairs.
> > | KD              | CE     |λ = 0.4 |λ = 1.0 | λ = 1.4    |λ = 2.0 |λ = 2.4 | λ = 3.0 | λ = 4.0 |
> > |-----------------|--------|--------|--------|------------|--------|--------|---------|---------|
> > | Teacher CE Loss | 0.560  | 0.568  | 0.568  | 0.582      | 0.630  | 0.595  | 0.627   | 0.871   |
> > | Teacher CMI     | 0.7180 | 0.7257 | 0.7268 | 0.7382     | 0.7869 | 0.7523 | 0.7825  | 0.9866  |
> > | Student Acc (%) | 70.660 | 71.362 | 71.452 | **71.654** | 71.274 | 71.370 | 71.290  | 70.694  |
> >
> >
> > >Q3. Did the authors observe any instability during training due to learning rate choice? How robust is DJIP to different optimizer configurations (e.g., step size, momentum, or learning)
> >
> > **Reply**: Thank you for the thoughtful question. As shown in **Table 9** of the revised manuscript, we trained the JPEG layer with various learning rates. A high learning rate leads to unstable updates, manifested by excessively large CMI values and noticeably degraded CE loss, which ultimately undermines the student’s performance. The learning rate of $0.01$ yields the best overall results, indicating that a stable yet sufficiently responsive update regime is essential for effective distillation.
> >
> > **Table 9**: Effect of learning rate (LR) in DJIP on ImageNet dataset and ResNet-34 $\rightarrow$ ResNet-18 pairs.
> > | KD              | CE     | LR = 0.1 | LR = 0.02 | LR = 0.01  | LR = 0.001 |
> > |-----------------|--------|----------|-----------|------------|------------|
> > | Teacher CE Loss | 0.560  | 0.875    | 0.601     | 0.582      | 0.561      |
> > | Teacher CMI     | 0.7180 | 0.9961   | 0.7577    | 0.7382     | 0.7191     |
> > | Student Acc (%) | 70.660 | 70.628   | 71.444    | **71.654** | 71.432     |
> >
> > For momentum and weight decay, we follow the implementation and setup from JPEG-DL (Salamah et al.) as discussed in Section A.5.1.
> >
> >
> > >Q4. Have you considered using other differentiable codecs (e.g., differentiable WebP or learned compression networks) as a generalization of DJIP?
> >
> > **Reply**: Thank you for the insightful question. As noted in our earlier response to Comment W1, the proposed alternating optimization framework is **fully agnostic** to the specific choice of input perturbation mechanism. In Theorem 1, the parameter set $\omega$ represents **any** learnable perturbation parameters and is not restricted to JPEG quantization tables. The differentiable JPEG module used in our experiments is simply one practical instantiation, selected for its simplicity, computational efficiency, and the availability of mature differentiable implementations.
> >
> > In principle, the perturbation layer can indeed be replaced by any differentiable codec, including differentiable WebP or learned compression networks. Consistent with this generality, Section A.7 demonstrates that replacing JPEG with alternative differentiable perturbation mechanisms—such as a **convolutional autoencoder**—while maintaining the same alternating optimization scheme yields comparable performance improvements. This confirms that DJIP is not tied to a specific codec but serves as a general optimization framework.
> >
> > **We thank you once again for the time and effort you have dedicated to reviewing our work. If our responses satisfactorily address all the concerns you raised, we would be grateful if you could consider raising the score to a higher level.**

---

### Official Review · Reviewer_JMaG · 2025-11-05

**Soundness:** 3
**Presentation:** 3
**Contribution:** 3
**Rating:** 6
**Confidence:** 4

**Summary:**

This paper proposes DJIP, a plug-and-play framework to enhance KD without modifying teacher model weights. Experiments show that this method has certain effectiveness. Experiments show that this method has certain effectiveness.

**Strengths:**

1. This paper improves the effectiveness of knowledge distillation without modifying the teacher model’s weights. The work is both interesting and effective.
2. The paper is well written, and the proposed method is plug-and-play and easy to follow.
3. The paper provides a detailed and thorough theoretical analysis.

**Weaknesses:**

1. The application scope demonstrated in this paper is somewhat limited. Could the proposed method be extended to LLMs (e.g., LLaMA) or generative models (e.g., Stable Diffusion)? Discussing applications beyond classification tasks would help enhance the paper’s breadth and contribution.
2. The JPEG-layer perturbation operates in the pixel space. It is unclear whether such pixel-level perturbations have a significant impact on the feature space. The rationality and reliability of the supervision signal constructed in this manner are therefore questionable.
3. The paper lacks an explanation regarding the selection strategy for key parameters, such as the frequency of alternating updates.
4. The paper does not sufficiently discuss several recent SOTA works[1,2,3,4,5,6] on knowledge distillation.

[1] f-Divergence Minimization for Sequence-Level Knowledge Distillation. ACL 2023.

[2] DistiLLM: Towards Streamlined Distillation for Large Language Models. ICML 2024.

[3] MiniLLM: Knowledge Distillation of Large Language Models. ICLR 2024.

[4] Rethinking Kullback-Leibler Divergence in Knowledge Distillation for Large Language Models. COLING 2025.

[5] ABKD: Pursuing a Proper Allocation of the Probability Mass in Knowledge Distillation via alpha-beta-Divergence. ICML 2025.

[6] DA-KD: Difficulty-Aware Knowledge Distillation for Efficient Large Language Models. ICML 2025.

**Questions:**

In addition to the issues mentioned in the Weaknesses, I have a few more concerns:

1. The paper claims orthogonality between DJIP and MCMI since DJIP explicitly optimizes the input space. Could the authors provide quantitative evidence (e.g., gradient cosine similarity between DJIP and MCMI objectives) to support this orthogonality claim?
2. Does increasing the number of quantization parameters consistently improve performance, or is there a saturation point?

---

> ### Author Response · Authors · 2025-11-25
> **Reply to Reviewer JMaG (part 1)**
>
> Dear Reviewer **JMaG**:
>
> We thank you for taking the time to review our paper and provide valuable feedback. Below, please find our responses to your comments.
>
>
> ## Summary
> >This paper proposes DJIP, a plug-and-play framework to enhance KD without modifying teacher model weights. Experiments show that this method has certain effectiveness. Experiments show that this method has certain effectiveness.
>
> **Reply**: We appreciate your summary of our contribution. We would like to clarify that, in addition to its plug-and-play property, a core technical contribution of DJIP is the proposed **alternating optimization algorithm**. This algorithm alternately updates the parameters of the input perturbation layer and the dummy “backward channel,” while keeping all pretrained model weights fixed. The differentiable JPEG module used in our experiments is only one instantiation of the input perturbation mechanism—chosen for its simplicity and efficiency—and can be replaced by any other differentiable codec, such as a **convolutional autoencoder**, as shown in Section A.7 of the revised manuscript. Thus, our contribution extends beyond the JPEG implementation and provides a general optimization framework for enhancing KD without modifying teacher model weights.
>
> ## Weaknesses
> > W1.  The application scope demonstrated in this paper is somewhat limited. Could the proposed method be extended to LLMs (e.g., LLaMA) or generative models (e.g., Stable Diffusion)? Discussing applications beyond classification tasks would help enhance the paper’s breadth and contribution.
>
> **Reply**: Thank you for raising this important point. Both JPEG and our DJIP framework are primarily designed for vision-based tasks, and we understand your interest in potential extensions to broader domains such as LLMs (e.g., LLaMA) or generative models (e.g., Stable Diffusion). Theoretically, DJIP has the potential to generalize beyond classification, since the core alternating optimization algorithm does not rely on image-specific assumptions. However, several non-trivial challenges remain unsolved for these domains.
>
> First, it is unclear how to define or construct _differentiable input perturbation mechanisms_ suitable for text or token sequences. Potential ways could be explored, but each presents non-trivial limitations. For example, one might consider (i) **perturbing continuous token embeddings** through adversarial or learnable noise (e.g., FGSM/PGD-style updates)[1, 2], (ii) **optimizing soft prompts or continuous prefix vectors** as surrogate perturbations [3, 4], or (iii) **using a learnable paraphrasing module** (e.g., a small seq2seq network [5]) to generate parametric, differentiable approximations of discrete token edits. While these techniques enable gradient-based updates, they do not translate cleanly to discrete token spaces and often break alignment between perturbed embeddings and actual text realizations. Moreover, unlike images, where perturbations remain within the same perceptual manifold, text perturbations frequently introduce semantic drift, length, or grammatical inconsistencies, making reliable and stable optimization substantially more difficult. As a result, extending DJIP to the textual domain would require additional theoretical and algorithmic innovations to preserve semantic fidelity while supporting end-to-end differentiability.
>
> Second, the definition of _conditional mutual information (CMI)_ and the construction of meaningful _cluster centroids_ for LLMs is substantially more complex than in vision tasks. In classification-based vision models, class labels naturally induce a well-defined partition of the representation space, enabling centroid estimation through supervised feature aggregation. This characteristic still exists in vision segmentation tasks. In contrast, LLMs operate on high-dimensional token trajectories and support a wide range of open-ended generation behaviors, making it unclear how to define discrete “clusters’’ that correspond to meaningful conditional variables. Possible proxies—such as grouping by **next-token distributions**[6], **semantic intent**[7], or **latent discourse states**[8]—remain unstable and highly sensitive to prompt structure. Furthermore, CMI in generative models depends on the joint distribution over entire output sequences rather than a fixed label, which complicates both its estimation and its maximization.
>
> These challenges imply that adapting DJIP to LLMs would require more insights into cluster structure and input perturbation methods. We regard the above directions as highly promising but beyond the scope of the current paper.
>
> **References:**
>
> [1] Sato, M., Suzuki, J., Shindo, H., & Matsumoto, Y. (2018). *Interpretable Adversarial Perturbation in Input Embedding Space for Text*. In _IJCAI-ECAI 2018_. arXiv:1805.02917.

---

> ### Author Response · Authors · 2025-11-25
> **Reply to Reviewer JMaG (part 2)**
>
> [2] Yuan, L., Zhang, Y., Chen, Y., & Wei, W. (2023). *Bridge the Gap Between CV and NLP! A Gradient-based Textual Adversarial Attack Framework*. In A. Rogers, J. Boyd-Graber, & N. Okazaki (Eds.), _Findings of the Association for Computational Linguistics: ACL 2023_, 7132–7146. Toronto, Canada: Association for Computational Linguistics.
>
> [3] Petrov, A., Torr, P. H. S., & Bibi, A. (2023). *When Do Prompting and Prefix-Tuning Work? A Theory of Capabilities and Limitations*. arXiv:2310.19698.
>
> [4] Li, Z., Liu, Y., Su, Y., & Collier, N. (2025). "Prompt Compression for Large Language Models: A Survey". In L. Chiruzzo, A. Ritter, & L. Wang (Eds.), _Proceedings of the 2025 Conference of the Nations of the Americas Chapter of the Association for Computational Linguistics: Human Language Technologies (NAACL 2025, Volume 1: Long Papers)_, 7182–7195. Albuquerque, New Mexico: Association for Computational Linguistics.
>
> [5] Chen, W., Tian, J., Xiao, L., He, H., & Jin, Y. (2020). *A Semantically Consistent and Syntactically Variational Encoder-Decoder Framework for Paraphrase Generation*. In D. Scott, N. Bel, & C. Zong (Eds.), _Proceedings of the 28th International Conference on Computational Linguistics (COLING 2020)_, 1186–1198. Barcelona, Spain (Online): International Committee on Computational Linguistics.
>
> [6] Zhao, Y., Behnia, T., Vakilian, V., & Thrampoulidis, C. (2024). "Implicit Geometry of Next-token Prediction: From Language Sparsity Patterns to Model Representations". arXiv:2408.15417.
>
> [7] Liu, J., Shang, Z., Ke, W., Wang, P., Luo, Z., Liu, J., Li, G., & Li, Y. (2025). "LLM-Guided Semantic-Aware Clustering for Topic Modeling". In W. Che, J. Nabende, E. Shutova, & M. T. Pilehvar (Eds.), _Proceedings of the 63rd Annual Meeting of the Association for Computational Linguistics (ACL 2025, Volume 1: Long Papers)_, 18420–18435. Vienna, Austria: Association for Computational Linguistics.
>
> [8] Saglam, B., Kassianik, P., Nelson, B., Weerawardhena, S., Singer, Y., & Karbasi, A. (2025). "Large Language Models Encode Semantics in Low-Dimensional Linear Subspaces". arXiv:2507.09709.
>
>
> >W2. The JPEG-layer perturbation operates in the pixel space. It is unclear whether such pixel-level perturbations have a significant impact on the feature space. The rationality and reliability of the supervision signal constructed in this manner are therefore questionable.
>
> **Reply**: To address this concern, we first recall _Proposition 1_ from MCMI (Ye et al., 2024), which has been formally proved in the appendix:
>
> "_Proposition 1_: If $\mathcal{f}_X$ is an intermediate feature map of a DNN corresponding to the input $X$, then $I(X; \hat{Y} \mid Y) \leq I(X; \mathcal{f}_X \mid Y)$."
>
> Based on this result, maximizing the CMI $I(X; \hat{Y} \mid Y)$ necessarily increases $I(X; \mathcal{f}_X \mid Y)$ as well. This implies that, regardless of the specific knowledge distillation paradigm—logit-based, feature-based, or relation-based—enhancing $I(X; \hat{Y} \mid Y)$ implicitly increases the upper bound of information propagated through the teacher’s feature hierarchy. Furthermore, DJIP is orthogonal to feature-based distillation methods (CC, RKD) that explicitly operate on intermediate representations. As a consequence, although the JPEG-layer perturbation is applied in the pixel domain, the resulting CMI-driven optimization objective still produces a substantial impact on the internal feature representations learned by the student network.
>
> >W3. The paper lacks an explanation regarding the selection strategy for key parameters, such as the frequency of alternating updates.
>
> **Reply**: Thank you for highlighting this omission. We indeed did not provide sufficient explanation of the selection strategy for this parameter in the original submission. In the revised version, we have added a dedicated discussion on the update interval (the number of mini-batches between two centroid updates), which now appears in **Table 10** of Section A.6 in the revised manuscript. This section also includes an ablation analysis illustrating how different interval choices affect optimization stability and final performance. Importantly, these results confirm that our originally reported setting remains the most effective among the tested configurations.

---

> ### Author Response · Authors · 2025-11-25
> **Reply to Reviewer JMaG (part 3)**
>
> **Table 10**: Effect of the centroid update interval in DJIP on CIFAR-100 dataset. Interval denotes the number of steps between two centroid updates.
> | Teacher   | Student | Method      | CMI   | KD    | DKD   | DIST  | CC    | RKD   |
> |-----------|---------|-------------|-------|-------|-------|-------|-------|-------|
> | ResNet-50 | VGG-8   | CE          | 0.009 | 73.81 | 73.94 | 74.11 | 70.25 | 71.50 |
> | ResNet-50 | VGG-8   | Interval=3  | 0.341 | 74.49 | 75.87 | 74.58 | 70.92 | 71.87 |
> | ResNet-50 | VGG-8   | Interval=5  | 0.341 | 74.48 | 75.87 | 74.80 | 70.92 | 71.93 |
> | ResNet-50 | VGG-8   | Interval=50 | 0.341 | 74.33 | 75.58 | 74.63 | 70.78 | 71.81 |
> | ResNet-50 | VGG-8   | Interval=500| 0.342 | 74.32 | 75.49 | 74.52 | 70.68 | 71.73 |
>
>
> >W4.  The paper does not sufficiently discuss several recent SOTA works[1,2,3,4,5,6] on knowledge distillation.
>
> **Reply**: Thank you for this helpful suggestion. In the revised manuscript, we have added a dedicated paragraph in Section A.1 of the revised manuscript discussing the recent state-of-the-art KD works [1–6] identified by you. We summarize their key ideas and highlight their relevance to our problem setting, clarifying how these studies provide conceptual insights for advancing KD in vision tasks. All corresponding references have been included in the updated appendix.
>
>
> ## Questions
> >Q1. The paper claims orthogonality between DJIP and MCMI since DJIP explicitly optimizes the input space. Could the authors provide quantitative evidence (e.g., gradient cosine similarity between DJIP and MCMI objectives) to support this orthogonality claim?
>
> **Reply**: We appreciate your insightful comment. We apologize for the earlier ambiguity. Our use of the term _“orthogonality”_ was not intended to imply literal gradient orthogonality between DJIP and MCMI. Instead, we meant that the two methods target _complementary dimensions_ of the distillation process: MCMI optimizes the teacher’s representation through model-space updates, whereas DJIP optimizes the input-space perturbation while keeping all model weights fixed. As a result, their effects are **additive rather than redundant**, which is empirically supported by the fact that combining DJIP with MCMI yields further performance gains (Tables 5 and 6 of the original manuscript). We have clarified this terminology in the revised manuscript to avoid misunderstanding.
>
>
> >Q2. Does increasing the number of quantization parameters consistently improve performance, or is there a saturation point?
>
> **Reply**: Thank you for the thoughtful question. The differentiable JPEG layer used in our work follows the JPEG standard, which provides exactly 128 trainable quantization parameters. To increase the parameter capacity and explore a richer perturbation space, several extensions are possible. One option is to allocate a separate quantization table for each 8×8 block. JPEG-DL also proposes a more practical alternative—_multi-round quantization_—where each round introduces an additional set of 128 parameters. This expands the parameter budget from 128 to 640 when using 5 rounds. We adopt this strategy for ImageNet due to its higher resolution and visual diversity.
>
> As reported in **Table 8** of the revised manuscript, the student’s performance improves as the number of quantization rounds increases, but the gains saturate once the number of rounds exceeds five. Hence, 5 rounds represent the saturation point in our experiments. In summary, enlarging the quantization parameter space (e.g., from 1 to 5 rounds) is beneficial under our alternating optimization scheme, but performance plateaus beyond that range.
>
> **Table 8**: Effect of the number of quantization rounds in DJIP on ImageNet dataset and ResNet-34 $\rightarrow$ ResNet-18 pairs.
> | KD              | CE     |round=1 |round=3 | round=5    | round=6  |
> |-----------------|--------|--------|--------|------------|----------|
> | Teacher CE Loss | 0.560  | 0.560  | 0.561  | 0.582      | 0.768    |
> | Teacher CMI     | 0.7180 | 0.7178 | 0.7185 | 0.7382     | 0.9097   |
> | Student Acc (%) | 70.660 | 71.278 | 71.610 | **71.654** | 70.960   |
>
> **We thank you once again for the time and effort you have dedicated to reviewing our work. If our responses satisfactorily address all the concerns you raised, we would be grateful if you could consider raising the score to a higher level.**

---

> > ### Comment · Reviewer_JMaG · 2025-11-26
> >
> > Thank you for the detailed response. I carefully read this rebuttal as well as the comments from other reviewers and the corresponding responses. I find these replies to be effective. However, I would like to raise a further concern related to W1:
> >
> > - In your reply to W1, you mention that constructing differentiable input perturbations for LLMs is very challenging because textual perturbations often introduce semantic drift or break token consistency. Have you considered a hybrid method that performs perturbations in the continuous embedding space while adding a semantic preservation regularizer? Would such a hybrid strategy potentially restore part of the feasibility of applying DJIP to language models?

---

> > > ### Author Response · Authors · 2025-11-26
> > > **Reply to Reviewer JMaG (part 4)**
> > >
> > > > In your reply to W1, you mention that constructing differentiable input perturbations for LLMs is very challenging because textual perturbations often introduce semantic drift or break token consistency. Have you considered a hybrid method that performs perturbations in the continuous embedding space while adding a semantic preservation regularizer? Would such a hybrid strategy potentially restore part of the feasibility of applying DJIP to language models?
> > >
> > > **Reply**: Thank you for your insightful comment again. The idea of using a hybrid strategy that perturbs continuous embedding representations while enforcing a semantic preservation regularizer is indeed **feasible**. Prior work on adversarial or regularized embedding perturbations (e.g., SMART[1] and FreeLB[2]) demonstrates that such hybrid strategies can introduce meaningful yet well-controlled variations to text representations.
> > >
> > > That said, the main difficulty of extending DJIP to LLMs lies in formulating a suitable CMI objective for LLMs. Unlike vision tasks, where CMI is naturally defined over discrete labels, language modeling lacks explicit categorical supervision; constructing an analogous mutual-information objective over extremely large vocabularies or autoregressive distributions remains challenging.
> > >
> > > Beyond these perturbation strategies, prompted by your question, we would like to highlight **another idea** drawn from the JPEG layer in our DJIP framework. As discussed in Section 3.3 of the original manuscript, the JPEG layer simulates a standard codec while replacing the non-differentiable quantizer $\mathcal{Q}_u$ with a differentiable soft quantizer $\mathcal{Q}_d$. This new quantizer introduces controlled perturbations via quantization error, allowing the model to receive structured and learnable distortions during training.
> > >
> > > A similar principle may be transferable to LLMs. **Quantization** is one of the most widely used techniques for compressing large language models, and modern low-bit quantization methods [3, 4] generally demonstrate that moderate quantization-induced perturbations can **preserve model accuracy** and **maintain semantic consistency** across common downstream tasks. Following this intuition, one could analogously place a differentiable quantizer after the input embedding stage (e.g., after token or positional embeddings), enabling $\mathcal{Q}_d$ to inject soft quantization noise into the embedding representations. Under our **alternating optimization** scheme, such a soft quantizer could in principle be trained to maximize the CMI objective, thereby providing a structured and differentiable perturbation mechanism analogous to the JPEG layer in DJIP.
> > >
> > > However, regardless of the specific perturbation formulation—whether embedding-level perturbations or differentiable quantization—the fundamental challenge persists: defining a principled and tractable CMI objective for LLMs. Addressing this theoretical gap is essential before DJIP can be meaningfully extended to LLMs.
> > >
> > > [1] Jiang, H., He, P., Chen, W., Liu, X., Gao, J., & Zhao, T. (2020). *SMART: Robust and Efficient Fine-Tuning for Pre-trained Natural Language Models through Principled Regularized Optimization*. In D. Jurafsky, J. Chai, N. Schluter, & J. Tetreault (Eds.), _Proceedings of the 58th Annual Meeting of the Association for Computational Linguistics_, 2177–2190. Online: Association for Computational Linguistics. doi:10.18653/v1/2020.acl-main.197.
> > >
> > > [2] Zhu, C., Cheng, Y., Gan, Z., Sun, S., Goldstein, T., & Liu, J. (2019). *FreeLB: Enhanced Adversarial Training for Natural Language Understanding*. arXiv:1909.11764.
> > >
> > > [3] van Baalen, M., Kuzmin, A., Koryakovskiy, I., Nagel, M., Couperus, P., Bastoul, C., Mahurin, E., Blankevoort, T., & Whatmough, P. (2024). *GPTVQ: The Blessing of Dimensionality for LLM Quantization*. arXiv:2402.15319.
> > >
> > > [4] Zhu, X., Li, J., Liu, Y., Ma, C., & Wang, W. (2024). *A Survey on Model Compression for Large Language Models*. Transactions of the Association for Computational Linguistics, 12, 1556–1577. Cambridge, MA: MIT Press. doi:10.1162/tacl_a_00704.

---

> > > > ### Comment · Reviewer_JMaG · 2025-11-27
> > > >
> > > > Thank you for your detailed and thoughtful responses. They effectively address my concerns and clarify the key points. Although extending the method to LLMs and diffusion models remains challenging, the explanations provided by the authors are insightful. I hope the authors can include all these clarifications in the next revision. I will slightly raise my score to accept.

---

> ### Author Response · Authors · 2025-11-27
> **Gratitude for the Reviewer JMaG's Insights**
>
> We sincerely thank you for the **thoughtful and constructive feedback**. Your insights have greatly enhanced the clarity and rigor of our work, and they have truly helped elevate the paper to **a higher level**. We deeply appreciate the time and expertise you invested in reviewing our manuscript, as well as your **supportive and encouraging decision**. All clarifications will be incorporated into the **camera-ready** version.

---

### Comment · Area_Chair_idcM · 2025-11-22

Dear Reviewers,

Thank you for your time and effort in reviewing submissions for ICLR  2026. As we begin the author-reviewer discussion process, we kindly remind you to submit your responses to the author rebuttals by **December  2**.


Your engagement in this discussion phase is crucial to ensuring a fair and thorough evaluation of each submission.

**Action Required**


- Carefully consider the authors’ rebuttal and any additional evidence they provide.

- Update your review (if applicable) to reflect your revised perspective.

-  **Discuss with the authors if further details are required**


Your AC

---

### Author Response · Authors · 2025-12-03
**Summary of Author Responses for Re-assigned AC**

Dear re-assigned AC,

We sincerely appreciate your additional reviewing effort under this year’s exceptional circumstances, as well as the continued dedication of all reviewers, ACs, SACs, and PCs. To help streamline your workload, we provide below a concise summary of our contributions and rebuttal efforts.

### **1. Main Contributions.**
Our **primary contribution** is a generic alternating optimization algorithm that iteratively updates a plug-and-play input-perturbation module and an auxiliary "backward channel" to maximize the CMI value of the teacher model, while keeping all teacher pretrained weights fixed. As a practical instantiation, we adopt a differentiable JPEG layer as the perturbation module, and refer to the resulting framework as _Differentiable JPEG-based Input Perturbation_ (DJIP).
**Empirically**, across diverse architectures (CNNs and ViTs), datasets (CIFAR-100 and ImageNet-1K) and 10+ distillers (e.g., DKD, LSKD), DJIP consistently yields distillation improvements up to 4.11%, while requiring substantially lower computational cost than other perturbation-based approaches (CKD, TALD).

### **2. Reviewer Comments & Our Responses.**
During the rebuttal period, we thoroughly addressed all reviewers’ comments. Two reviewers explicitly acknowledged that our replies were effective and helpful: one increased the score to _8 (Accept)_, and another maintained  _6_. As the remaining two did not provide further responses, we summarize the **major comments** raised and our corresponding resolutions below.

> 1.  **Is DJIP tied to JPEG?**
>
>     No. The alternating optimization algorithm is independent of JPEG and compatible with any differentiable perturbation module. JPEG is merely one instantiation, selected for simplicity, efficiency, and mature differentiable implementations. Additional results using a convolutional autoencoder in _Table 11_ confirm this generality.
> 2.  **Hyperparameter sensitivity.**
>
>     Comprehensive ablation studies (_Table 7-10_) demonstrate the robustness to all key hyperparameters (e.g., **$\lambda$**, learning rate) and confirm the effectiveness of the selected settings.
> 3.  **Missing runtime, peak memory, and scaling analysis.**
>
>     We added measurements of throughput, runtime, and peak GPU memory (_Table 12 and C_), as well as scaling results over varying input resolutions and batch sizes (_Table A and B_), showing that DJIP introduces minimal computational overhead (only 128 additional parameters) and scales efficiently.
> 4.  **Convergence and stability discussion.**
>
>     _Section A.4_ now includes a formal theoretical analysis of convergence, supplemented by empirical verification on both small and large models. In our replies, we also elaborated on several structural properties of the DJIP formulation—such as the fixed teacher, the construction of the centroid mini-dataset, and the smoothing effects of EMA updates—that collectively contribute to stable training dynamics.
> 5.  **Experiments in low-data regimes.**
>
>     Few-shot KD results (_Table 13_) show that DJIP’s benefits become even more pronounced  (up to 5.81%) when data is scarce.
> 6.  **Why and how input perturbation improves distillation.**
>
>     _Figures 6 and 7_ visualize how perturbations affect the teacher’s output distributions over training. According to the conclusion in the original MCMI paper, maximizing the CMI drives the teacher’s predictions closer to the Bayes conditional probability distribution (BCPD), which is known to yield stronger distillation signals. This provides a principled explanation for why DJIP—by explicitly maximizing perturbed CMI—achieves improved distillation performance. Eigen-CAM visualizations in MCMI also confirm that teacher with higher CMI can provide richer “dark knowledge” to the student, consistent with many recent KD findings.
> 7.  **Extension beyond vision (e.g., LLMs).**
>
>     _Section A.1_ now includes a discussion on KD for LLMs. Our replies to reviewers also outline the challenges of extending DJIP to the textual domain: while the alternating optimization algorithm is generic, the lack of suitable differentiable perturbation mechanisms and the difficulty of defining cluster structures and CMI for discrete text make such extensions nontrivial. Nevertheless, If these challenges are resolved, our algorithm can in principle be applied to text-based tasks as well.

All other **minor comments** were addressed one by one and are fully resolved as well.

**In summary**, our work introduces a novel and practically useful algorithmic framework, offers strong empirical results, and provides insights of value to the ICLR community. Given that all comments and suggestions have been fully addressed through additional analyses and experiments, we would be extremely grateful if you could consider accepting this paper.

Thank you,
The Authors

---

### Meta-Review · Area_Chair_x6Fe · 2025-12-28

**Summary:**

The authors have basically addressed all concerns raised and actually responded comprehensively. There are no outstanding concerns remaining. I recommend accepting the paper.

**Reviewer Concerns:**

* Reviewer JMaG: The authors have addressed all concerns.

* Reviewer tbBh: The authors have addressed all concerns.

* Reviewer RUYs: The authors have addressed all concerns.

* Reviewer gUa4: The authors have addressed all concerns.

**Reviewer Scores:**

* Reviewer JMaG: I think the reviewer would have increased the score.

* Reviewer tbBh: I think the reviewer would have increased the score.

* Reviewer RUYs: I do not think the reviewer would have increased the score any further.

* Reviewer gUa4: I do not think the reviewer would have increased the score any further.

---

### Decision · Program_Chairs · 2026-01-26

Accept (Poster)